# Spatial transcriptomics reveals distinct and conserved tumor core and edge architectures that predict survival and targeted therapy response

Rohit Arora[1,16], Christian Cao[1,2,16], Mehul Kumar[1,3], Sarthak Sinha [4], Ayan Chanda [1,3], Reid McNeil[1,3], Divya Samuel[1,3], Rahul K. Arora [5,6], T. Wayne Matthews[7,8], Shamir Chandarana[7,8], Robert Hart[7,8], Joseph C. Dort[3,7,8,9], Jeff Biernaskie[4,10,11,12], Paola Neri[3,13], Martin D. Hyrcza[3,14] & Pinaki Bose [1,3,6,15] ✉

The spatial organization of the tumor microenvironment has a profound impact on biology and therapy response. Here, we perform an integrative single-cell and spatial transcriptomic analysis on HPV-negative oral squamous cell carcinoma (OSCC) to comprehensively characterize malignant cells in tumor core (TC) and leading edge (LE) transcriptional architectures. We show that the TC and LE are characterized by unique transcriptional profiles, neighboring cellular compositions, and ligand-receptor interactions. We demonstrate that the gene expression profile associated with the LE is conserved across different cancers while the TC is tissue specific, highlighting common mechanisms underlying tumor progression and invasion. Additionally, we find our LE gene signature is associated with worse clinical outcomes while TC gene signature is associated with improved prognosis across multiple cancer types. Finally, using an in silico modeling approach, we describe spatially-regulated patterns of cell development in OSCC that are predictably associated with drug response. Our work provides pan-cancer insights into TC and LE biology and interactive spatial atlases (http://www.pboselab.ca/spatial_OSCC/; http://www.pboselab.ca/dynamo_OSCC/) that can be foundational for developing novel targeted therapies.

[1]Department of Biochemistry & Molecular Biology, Cumming School of Medicine, University of Calgary, Calgary, AB, Canada. [2]Temerty Faculty of Medicine, University of Toronto, Toronto, ON, Canada. [3]Arnie Charbonneau Cancer Institute, Cumming School of Medicine, University of Calgary, Calgary, AB, Canada. [4]Department of Comparative Biology and Experimental Medicine, Faculty of Veterinary Medicine, University of Calgary, Calgary, AB, Canada. [5]Center for Health Informatics, University of Calgary, Calgary, AB, Canada. [6]Institute of Biomedical Engineering, University of Oxford, Oxford, United Kingdom. [7]Ohlson Research Initiative, Cumming School of Medicine, University of Calgary, Calgary, AB, Canada. [8]Section of Otolaryngology Head & Neck Surgery, Department of Surgery, Cumming School of Medicine, University of Calgary, Calgary, AB, Canada. [9]Department of Community Health Sciences, Cumming School of Medicine, University of Calgary, Calgary, AB, Canada. [10]Alberta Children's Hospital Research Institute, Cumming School of Medicine, University of Calgary, Calgary, AB, Canada. [11]Hotchkiss Brain Institute, Cumming School of Medicine, University of Calgary, Calgary, AB, Canada. [12]Department of Surgery, Cumming School of Medicine, University of Calgary, Calgary, AB, Canada. [13]Division of Hematology, Department of Oncology, University of Calgary, Calgary, AB, Canada. [14]Department of Pathology and Laboratory Medicine, University of Calgary, Calgary, AB, Canada. [15]Department of Oncology, Cumming School of Medicine, University of Calgary, Calgary, AB, Canada. [16]These authors contributed equally: Rohit Arora, Christian Cao. ✉e-mail: pbose@ucalgary.ca

Oral squamous cell carcinoma (OSCC) is the most common head and neck cancer and accounts for over 90% of cancers that develop in the mucosal epithelium of the oral cavity[1–3]. In countries such as India, OSCC is the most commonly diagnosed cancer[4]. Several carcinogenic risk factors, including tobacco and alcohol use, and HPV infection are associated with oral carcinogenesis[5]. Unlike other head and neck cancer subsites such as the oropharynx, HPV accounts for only 2–5% of OSCC and the significance of HPV infection in OSCC is currently unknown[6]. HPV-negative disease, mostly driven by tobacco and alcohol use, accounts for a majority of OSCC cases and improving the prognosis of this subset is an area of unmet need[5]. Despite advances in the understanding of OSCC biology over the past few decades, patient outcomes have remained largely static; less than 50% of HPV-negative OSCC patients survive more than 5 years[2]. Conventional treatment modalities such as surgery and cytotoxic chemotherapy have also yielded limited success and can result in severe morbidity[7,8], highlighting the need for alternative treatment strategies that are based on biologic insights.

OSCC invasion and metastasis is poorly understood and accounts for a majority of cancer-associated deaths[9]. More than 50% of patients experience locoregional recurrence or develop metastases within 3 years of treatment[9]. The OSCC leading edge (LE), comprised of tumor cell layers at the border of the OSCC tumor, has been previously identified to have prognostic value in clinical grading and may mediate tumor invasion and metastasis[10]. However, the active mechanisms at the invasive edge, and other important spatially-defined regions of carcinomas, are not fully understood[11]. Previous immunohistochemistry and in-situ hybridization efforts to study the LE have been limited to low-throughput analysis and have struggled to comprehensively characterize the OSCC tumor microenvironment[10,12].

Recent advances in single-cell RNA sequencing (scRNA-seq) have enabled the exploration of intratumoral heterogeneity in head and neck squamous cell carcinoma (HNSCC). For instance, tumor cells expressing a partial epithelial mesenchymal transition (p-EMT) program were localized at the LE and demonstrated enhanced invasive potential[13]. Conversely, tumor cells lacking p-EMT expression but expressing epithelial differentiation program markers were localized to the tumor core (TC)[13]. However, scRNA-seq studies ultimately lack the spatial information required to correlate transcriptional state dynamics with tumor topography[14]. Spatial transcriptomics (ST) builds upon scRNA-seq by providing expression data while simultaneously preserving the 2D positional information of cells, providing a holistic account of transcriptional heterogeneity in the tumor microenvironment[15].

Here, we leverage ST and single-cell RNAseq to unravel intratumoral transcriptional heterogeneity in OSCC by determining and characterizing the unique geographical regions of the OSCC tumor architecture. We find that malignant cells residing within the TC and LE possess unique transcriptomic profiles and ligand receptor interactions, which may be explained by the presence of spatially unique cancer cell states. We also discover that the conserved transcriptional programs in the TC and LE have prognostic value not only in OSCC, but across multiple cancer types. Using predictive machine learning models, we observe that the LE regions acquire conserved features shared across cancer types, while the TC is more cancer-type specific. Furthermore, our work leverages RNA velocity inference to identify patterns of tumor differentiation within the TC and LE. Finally, using *in-silico* modeling, we identify potentially effective drugs that disrupt the information flow from the TC to LE in OSCC patients. Together, our results provide insights into the complex OSCC landscape and indicate that solid tumors may employ a conserved and common set of mechanisms to progress and invade that may be targeted for therapeutic benefit.

## Results

### ST profiling and cellular deconvolution of OSCC samples

We performed ST on 12 fresh-frozen surgically resected OSCC samples from 10 unique patients using the 10x Genomics Visium platform (Fig. 1a and Supplementary Table 1). Transcriptomes from 24,876 spots were sequenced to 43,648 post-normalization mean reads per spot. Data was normalized, corrected for batch effects across samples and dimensionality reduced for subsequent analysis. Hematoxylin and Eosin (H&E)-stained images from each tumor sample were examined and morphological regions were annotated by the study pathologist (M.H.) (Fig. 1b and Supplementary Fig. 1a–m).

We next determined the composition of malignant tumor cells and other cellular subpopulations present in the pathologist-annotated squamous cell carcinoma regions by performing integrative analysis of our ST data with a separate, publicly-available HNSCC scRNA-seq dataset[13]. To identify malignant tumor spots, we stringently characterized malignant cells as having a deconvolution score >0.99 (Fig. 1c), or CNV probability score >0.99 (Fig. 1d). CNV analysis revealed recurrent deletions in chromosome 3, and amplifications in chromosome 9 (Supplementary Fig. 1n). All 12 samples were identified to have both spatially deconvolved or CNV-inferred cancer cells based on the applied cutoff with high confidence, resulting in 13950 malignant and 10852 nonmalignant spots (Fig. 1e and Supplementary Fig. 1o). Following batch effect correction and dimensionality reduction of our classified cells, uniform manifold approximation and projections (UMAP) of ST spots showed a scattered distribution of malignant spots, reflecting a continuum of transcriptional profiles (Fig. 1e). CAF subtypes conserved in HNSCC were annotated using the marker genes *LRRC15* and *GBJ2* for ecm-MYCAFs, and *ADH1B* and *GPX3* for detox-iCAFs (Fig. 1f)[16,17]. Spatial deconvolution analysis revealed that cancer, ecm-myCAF, intermediate fibroblasts, detox-iCAF, dendritic, mast, macrophage, and cytotoxic CD8+ T cell types were present in nearly all samples (Fig. 1f and Supplementary Fig. 1p).

### Unsupervised clustering reveals that the TC and LE are functionally heterogeneous components of the tumor microenvironment

After identifying and annotating malignant tumor spots that were primarily composed of cancer cells, we performed unsupervised louvain-clustering to unravel the spatial heterogeneity in cancer cell expression profiles. We generated 14 louvain clusters among aggregated malignant spots that could be partitioned into 3 major clusters (Fig. 2a). We then characterized the major clusters through differential gene expression analysis (DGEA) (Fig. 2b). Top DEGs enriched in cluster 1 included genes involved in keratinization *SPRR2D*, *SPRR2E*, *SPRR2A*, and inhibition of EMT *DEFB4A* and *LCN2* (ref. [18,19]), while DEGs in cluster 3 included genes involved in the ECM matrix *COL1A1*, *FN1*, *COL1A2*, *TIMP1*, *COL6A2* (Fig. 2b and Supplementary Data 1). DEGs enriched in cluster 2 shared attributes of both cluster 1 and 3, with genes involved in keratinization *KRT6C*, *KRTDAP*, *KRT6B* (ref. [20]), and ECM remodeling *LYPD3*, *SLPI* (refs. [21,22]) (Fig. 2b and Supplementary Data 1). Interestingly, the expression of *CLDN4* and *SPRR1B* HNSCC TC markers[13], and *LAMC2* and *ITGA5* HNSCC LE markers[13] corresponded to clusters 1 and 3, respectively (Fig. 2b, c and Supplementary Data 1). These findings prompted us to annotate cluster 1 as "tumor core" (TC) and cluster 3 as "leading edge" (LE). Cluster 2 was annotated as "transitory" due its composition of TC and LE DEG programs (Fig. 2d and Supplementary Fig. 2a–m).

We then sought to determine whether the patterns of gene expression in the LE and TC were conserved across different patients. To do this, a correlation matrix was generated from the whole transcriptome gene expression profiles within the two spatial regions (Fig. 2e). A high degree of correlation was generally observed within the TC, and within the LE, across different patients. Interestingly, the

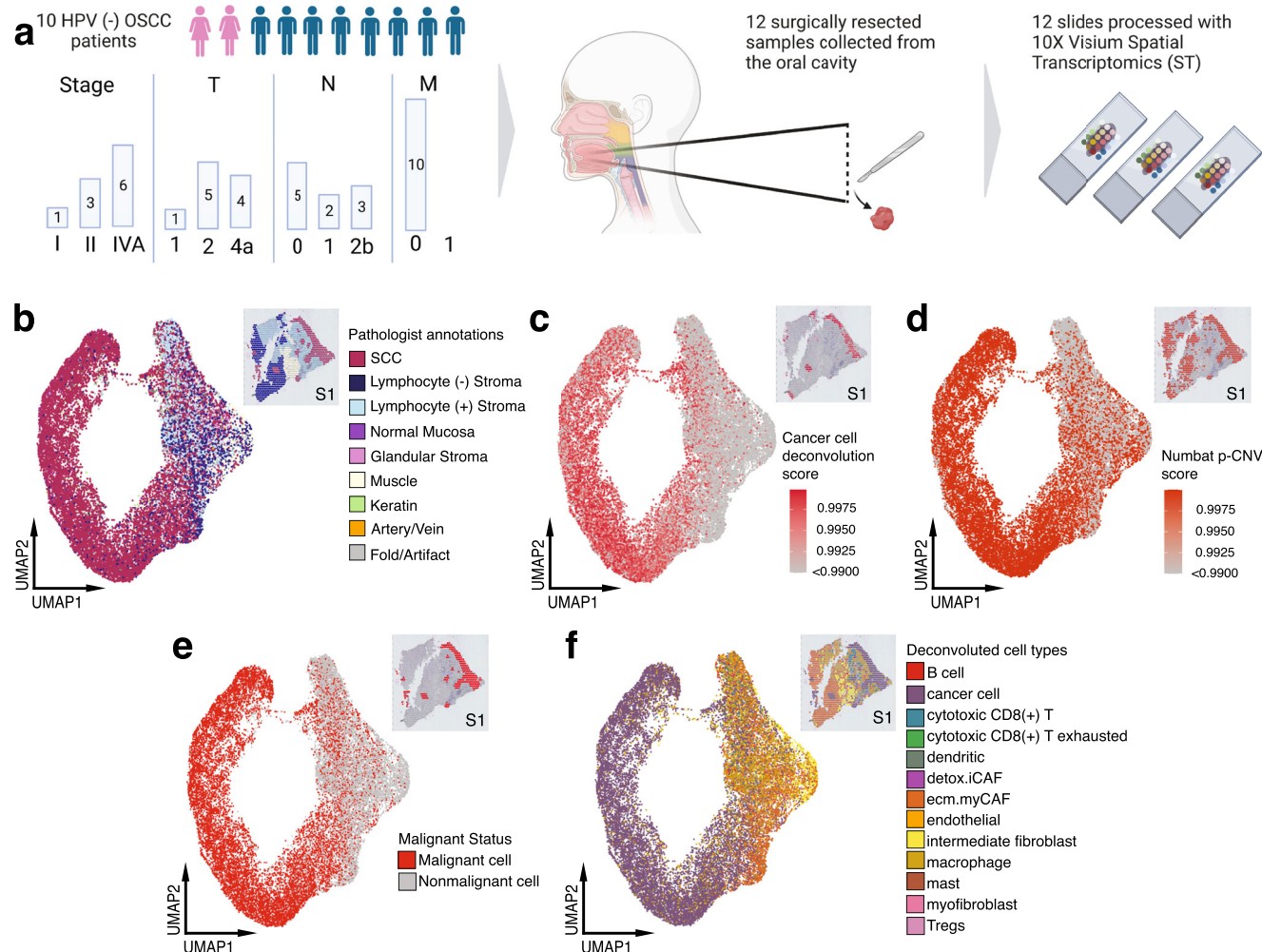

**Fig. 1 | Overview of experimental design for ST analysis and cellular deconvolution of OSCC patient samples. a** Schematic representing patient clinical data and sample acquisition and processing strategy. UMAP projection of 24,876 spots aggregated from all 12 spatially-profiled samples colored based-on. Created with BioRender. **b** Pathologist annotations, **c** single-cell HNSCC deconvolution based on scRNA-seq data from Puram et al., **d** CNV probability per spot, **e** malignant spot status, and **f**. spot annotations based on deconvolution and CNV probabilities. HPV human papillomavirus, OSCC oral squamous cell carcinoma, SCC squamous cell carcinoma, Tregs T-regulatory cell.

correlation between the TC and LE expression programs within each patient was relatively low, highlighting the distinct nature of these compartments in the tumor microenvironment. Therefore, our TC and LE annotations represent regions with unique transcriptomic profiles that are conserved across patients.

To explore the functional differences between TC and LE spots, we queried cancer hallmark and key oncogenic pathway-associated gene-sets for their expression within these compartments. We found that LE spots displayed higher expression of genes associated with cell cycle (*p*-adj < 0.001), epithelial-mesenchymal transition (EMT) (*p*-adj < 0.05), and angiogenesis (*p*-adj < 0.001) (Supplementary Fig. 2n). EMT scores in the LE were also expressed over a broad range (Supplementary Fig. 2n), in agreement with the EMT continuum model[11]. We additionally queried a published epithelial differentiation and p-EMT gene-set and observed localization of these programs to TC and LE spots, respectively (*p*-adj < 0.0001, *p*-adj < 0.001) (Supplementary Fig. 2n)[13]. Cellular function hallmarks that were upregulated in the TC included keratinization, cell differentiation, as well as antimicrobial and immune-related pathways, while protein translation and ribosome-related pathways were upregulated in the LE (Supplementary Fig. 2o and Supplementary Data 2). Ingenuity Pathway Analysis (IPA) predicted the activation of GP6, EIF2, and HOTAIR regulatory canonical signaling pathways

exclusively in the LE across patients (Fig. 2f)[23–25]. These characteristics might reflect the role of the OSCC LE in governing invasive and metastatic behavior. In the TC, we observed the activation of MSP-RON signaling in macrophages, IL-33, and p38 MAPK canonical signaling pathways, as well as downregulation of LXR/RXR and SPINK1 canonical signaling pathways across most patients (Fig. 2f). These findings suggest that the OSCC TC may modulate the immune response within the tumor microenvironment, and has a role in promoting or inhibiting tumor progression[26–30].

We next explored regulatory differences between the TC and LE using single-cell regulatory network inference and clustering (SCENIC) to infer transcription factor (TF) activity. SCENIC analysis identified the upregulation of several proto-oncogenic TFs *EGR3* and *DLX5* (refs. 31,32), and tumor suppressor TFs *MXI1*, *GRHL3*, and *PITX1* (refs. 33–35) in the TC (Supplementary Fig. 2p and Supplementary Data 3). Conversely, the upregulation of several TFs including cellular development and differentiation-regulatory genes *TP63* and *HOXB2* (refs. 36,37), and EMT regulatory genes *CREB3L1*, *TCF4*, and *NFATC4* (refs. 38–40) were observed in the LE (Supplementary Fig. 2p and Supplementary Data 2). IPA upstream regulatory analysis corroborated the activation of several proto-oncogenic TFs *EHF* and *BCL3* in the TC (refs. 41,42), and EMT regulatory genes *SORL1* and EGFR in the LE (Supplementary Fig. 2q) (refs. 43,44).

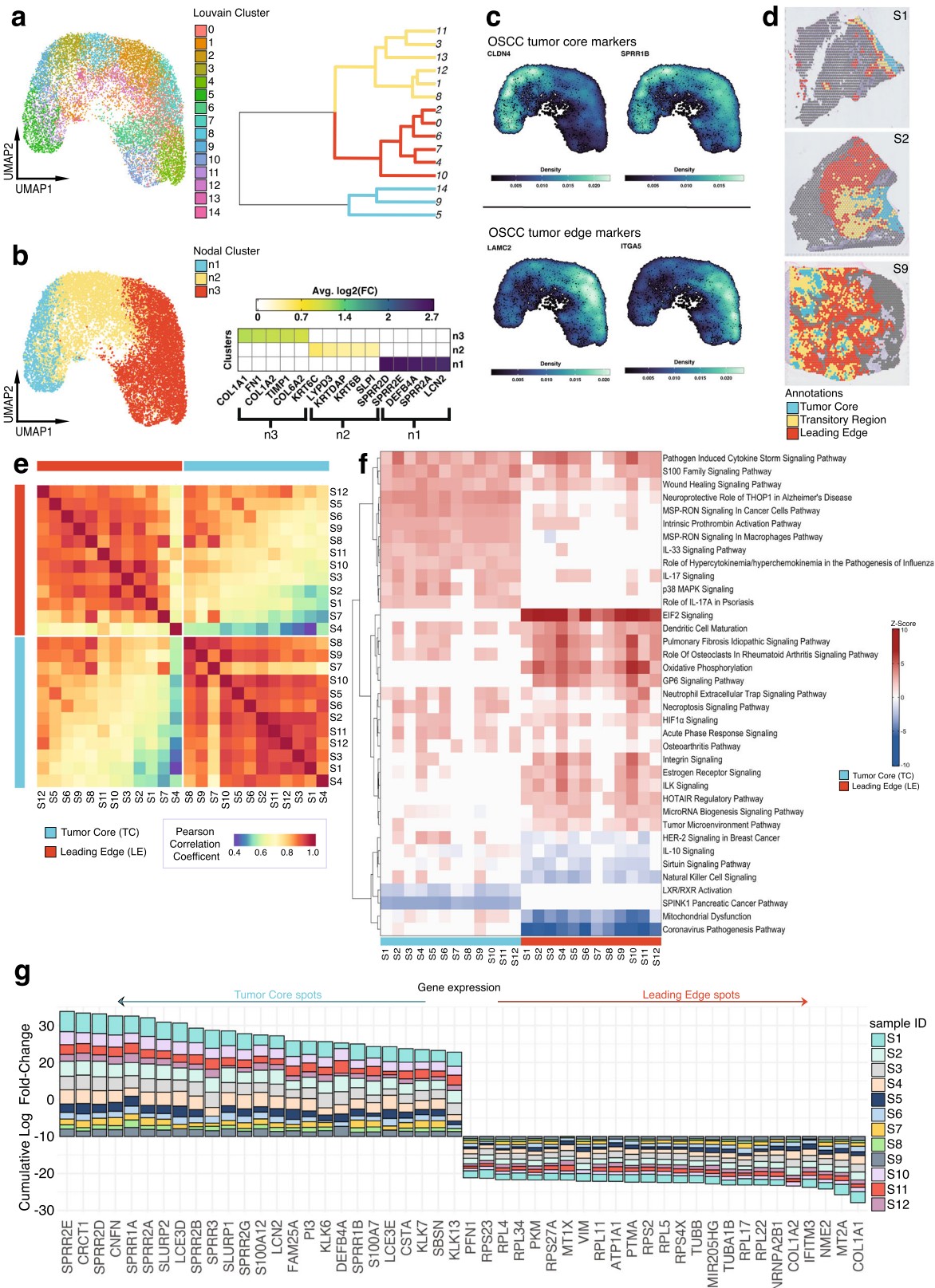

Differential expression (DE) analysis between the TC and LE revealed 117 genes upregulated in the TC and 91 in the LE, across 10 or more samples (Fig. 2g and Supplementary Data 3). Upon comparison to a previous HNSCC scRNA-seq study[13], only 40 epithelial differentiation and 7 p-EMT DEGs overlapped with our TC and LE DEGs, respectively. Top differentially expressed genes in the TC included genes involved in keratinization *SPRR2E*, *CRCT1*, *SPRR2D*, *CNFN*, and *SPRR1A*, while top genes in the LE included collagens (*COL1A1*, *COL1A2*), and genes involved in EMT initiation and regulation *MT2A*, *NME2*, *IFITM3* (refs. 45–48), highlighting the presence of a fibrovascular niche (Fig. 2g and Supplementary Data 4).

**Fig. 2 | TC and LE are spatially unique regions in the OSCC microenvironment.**
**a** UMAP projection of 13950 malignant spots aggregated from all 12 spatially-profiled samples, partitioned by Louvain clusters with an accompanying phylogenetic tree demonstrating cluster transcriptomic similarity. **b** UMAP projection of 13950 malignant spots aggregated from all 12 spatially-profiled samples partitioned by three major nodal clusters, with an accompanying heatmap visualizing the log$_2$(FC) of the top 5 DEGs for each cluster. **c** Nebulosa kernel density plot visualizing gene expression of literature-validated OSCC tumor core and leading edge markers. **d** TC transitory, and LE annotations for samples 1, 2, and 9. **e** Whole transcriptome Pearson correlation heatmap of TC and LE annotations across all

spatially-profiled samples. Samples are ordered based on transcriptomic similarity. **f** Ingenuity Pathway Analysis heatmap visualizing predicted activation and deactivation of TC and LE pathways. Pathways are displayed if they are activated or deactivated across 10 or more samples and ordered based on similarity of z-score for each pathway across samples. **g** Consensus plot displaying the cumulative average logFC for the top 25 genes significantly differentially expressed between the TC and LE across more than 9 samples (adj. $p < 0.001$, two-sided Wilcox rank sum test, Bonferroni correction). Source data are provided as a Source Data file where relevant. UMAP uniform manifold approximation projection, OSCC oral squamous cell carcinoma.

## Distinct cancer cell states inhabit the TC and LE

Four HNSCC molecular subtypes classified by patterns of gene expression and clinical outcomes have been previously identified and validated by the The Cancer Genome Atlas (TCGA)[49,50]. To determine if the unique expression profiles observed in the OSCC TC and LE could be attributed to the composition of HNSCC molecular subtypes, we integrated TCGA subtype expression signatures with our ST dataset. Spots within the TC and LE were scored for their correspondence to subtype expression signatures. We found that multiple molecular subtypes may be present within the same tumor microenvironment across several patients, with no consistent pattern of subtype composition (Supplementary Fig. 3a). Overall, the TC state was most enriched for the Basal subtype ($p < 0.0001$) (Supplementary Fig. 3b), while the LE was most depleted for the atypical subtype ($p < 0.01$) (Supplementary Fig. 3c). Therefore, strictly classifying patients into gene expression-based subtypes in HNSCC may result in over-simplification of the complex biology of these cancers. Next, we considered the association between tumor subclonal architectures and the OSCC TC and LE by inferring clonal lineages and evolutionary history through CNV events with the Numbat package. We found that multiple subclonal lineages were present throughout the OSCC tumor, with similar proportions of subclonal populations across TC and LE regions (Supplementary Fig. 3d). These findings imply that the transcriptomic differences between the TC and LE cannot be entirely explained by intratumoral genetic diversity.

We then asked if the contribution of cancer stem cells (CSCs) could help explain the differences in TC and LE expression profiles. CSCs are cancer cell populations that possess stem-cell like progenitor and malignant properties[51]. Given the abundance of EMT-related, metastatic, and invasive expression programs at the LE, we hypothesized that CSCs may be preferentially localized in the LE. However, we found no significant differences in the expression of canonical OSCC CSC markers[52] between the LE and TC ($p > 0.05$) (Fig. 3a). Furthermore, expression of CSC markers were seen evenly throughout UMAP projections, indicating that CSC populations are found throughout the OSCC tumor (Fig. 3a; density plot) along with non-stem-like malignant cells.

Therefore, we believe that the distinct biological profiles of the TC and LE are explained by the presence of unique cancer cell states–conserved gene expression programs that dynamically manifest from specific tumor microenvironment interactions–comprising both CSC and non-stem-like malignant cells[53,54]. Previous literature exploring dynamic CSC states has proposed the existence of mesenchymal-like CSCs that inhabit the LE and epithelial-like CSCs inhabit the TC (Fig. 3b)[55]. When we scored gene-sets associated with these distinct CSC states to our ST dataset, our results corroborated the existence of higher expression of the mesenchymal-like CSC state in the LE ($p < 0.001$) and epithelial-like CSC state in the TC ($p < 0.001$) (Fig. 3c, d). The localization of these CSC states was further validated through immunofluorescence staining of serial tissue sections, which revealed localization of the CD24 marker at the TC, and the CD44 marker at the LE (Supplementary Fig. 3e). These findings reinforce the plasticity within the TC and LE niches that promote the propagation of transcriptionally unique cancer cell states.

## The TC and LE architectures display distinct ligand-receptor interactions

Given the considerably diverse spatial architecture of the OSCC TC and LE, we sought to elucidate the nature and role of intracellular and extracellular signaling interactions in the TC and LE. We performed cell-cell communication analysis with the CellChat package to derive quantitative inferences of intercellular communication networks (Supplementary Data 5). ANGPTL, GRN, NECTIN, and EPHB signaling pathways were exclusively seen in the TC, and CSPG4 in the LE (Fig. 3e and Supplementary Fig. 3f). Several ECM remodeling pathways including Collagen, Tenascin, and Laminin were also expressed in both the TC and LE (Fig. 3e and Supplementary Fig. 3f). Outgoing signaling patterns upregulated in LE-LE signaling, relative to TC-TC signaling, included Collagen, Laminin, Tenascin, FN1, MIF, APP, CD99, Notch, and CSPG4 pathways (Supplementary Fig. 3f), reinforcing the role of these pathways in facilitating cancer invasion and metastasis.

We next examined cell-cell communication mechanisms among malignant cells. We included ecm-myCAFs in this analysis because of their high cellular contribution across all tumor samples (Supplementary Fig. 3g). We found ecm-myCAFs exhibited prominent cellular signaling, with many more interactions to neighboring LE cancer cells compared to TC-TC and LE-LE signaling (Supplementary Fig. 3g). Moreover, ecm-myCAF-LE interaction strength greatly exceeded the interaction strength of LE-LE and TC-TC cancer cell signaling (Supplementary Fig. 3g). The greater number of ecm-myCAF-LE interactions and strength relative to LE-LE signaling highlights a critical role for ecm-myCAFs in modulating cancer cell behavior at the LE.

We then explored specific ligand-receptor pairs representative of TC-TC cancer cell, LE-LE cancer cell, and ecm-myCAFs-LE cancer cell signaling. We found that TC-TC cancer cell signaling could occur through adhesive ligand-receptor pairs *DSC2-DSG1* and *ANGPTL4-SDC1*, among others (Fig. 3f and Supplementary Data 5). Interestingly the *ANGPTL4-SDC1* pair may inhibit Wnt signaling[56], which is a major pathway implicated in cancer metastasis[57]. Similarly, LE-LE cancer cells could signal through adhesive ligand-receptor pairs *LAMB3-ITGA6_ITGB4* and *LAMB3-ITGA6_ITGB1*, and inflammatory ligand-receptor pairs *MIF-CD74_CD44* (Fig. 3g and Supplementary Data 5). The *MIF-CD74* ligand-receptor pair has been previously identified to initiate oncogenic signaling pathways[58]. Due to the high number of ecm-myCAF-LE cancer cell interactions, we also analyzed these signaling pathways. Our analysis strongly predicted adhesive *COL1A1-SDC1* and *FN1-SDC1* ligand-receptor interactions, among others (Fig. 3h and Supplementary Data 5).

To further characterize the tumor microenvironment, we identified and quantified the number of adjacent nonmalignant spots neighboring our malignant TC and LE spots. Nonmalignant neighboring spots were approximated as a specific cell type based on the most enriched non-cancer cell after deconvolution. Our analysis found significantly higher numbers of neighboring spots enriched for cytotoxic CD8(+) T cell ($p$-adj $< 0.01$), ecm.myCAF ($p$-adj $< 0.001$), intermediate fibroblast ($p$-adj $< 0.01$), and macrophage cells ($p$-adj $< 0.01$), neighboring LE spots, relative to TC spots

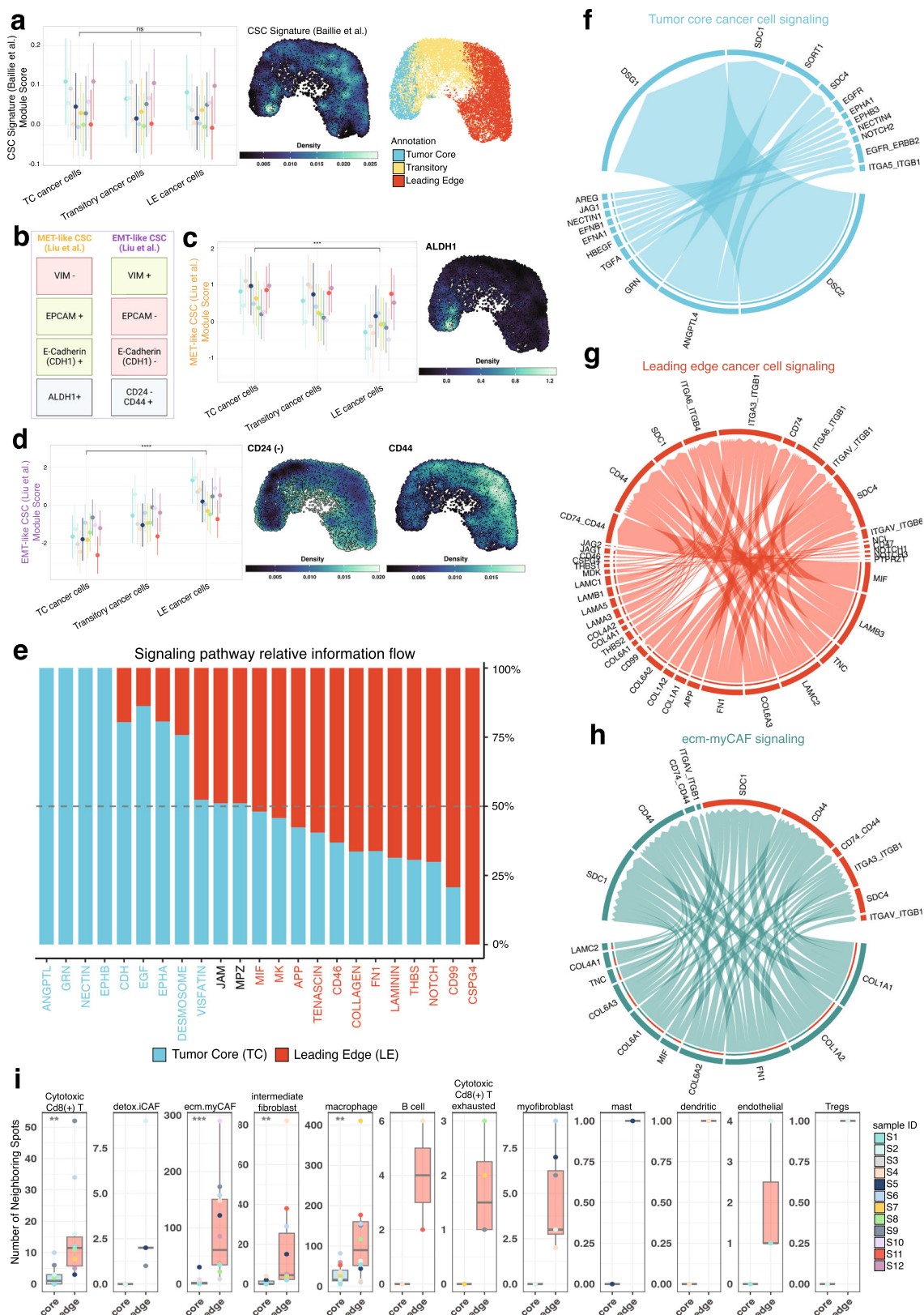

(Fig. 3i). Macrophages were the most abundant TC neighboring cells (Fig. 3i), and were found to contribute to adhesive desmosome and cadherin (CDH) signaling pathways with the TC (Supplementary Fig. 3h). Macrophage and cytotoxic CD8(+) T cells also participated in intercellular signaling with LE and ecm.myCAF cell types via

laminin and collagen signaling pathways (Supplementary Fig. 3i), highlighting their prominent role in cancer signaling. Taken together, our findings suggest that malignant cells in the TC and LE engage in distinct patterns of cell-cell communication, further shaping their unique biologies.

**Fig. 3 | TC and LE cancer cell states are distinct entities with unique ligand-receptor interactions. a** Comparative expression of a CSC gene signature across TC, LE, and other squamous cell carcinoma (SCC) spots, visualized with nebulosa kernel density plot. UMAP of TC, LE, and transitory cancer spot annotations are provided for spatial reference. Circles are representative of mean and lines represent standard deviation (*n* = 12 samples across 10 independent patients). **b** Schematic representation of epithelial (eCSC) and mesenchymal cancer stem cell (mCSC) markers characterized by Liu et al. [55]. **c, d** Comparative expression of mCSCs (*p*-value = 2e−04) and eCSC (*p*-value = 1.5e−06) gene sets across the TC, LE, and other SCC spots with nebulosa kernel density plot visualizing mCSCs and eCSC markers. Circles are representative of mean and lines represent standard deviation (*n* = 12 biologically independent samples). **e** Stacked bar plot visualizing enriched signaling pathways by overall information flow across TC and LE spots. Pathways are colored by their dominance within the TC or LE. Circos plots describing spatially deconvoluted ligand-receptor pairs involved in cell-cell interactions between

**f** cancer cells in the TC, **g** cancer cells in the LE, and **h** ecm-myCAF and cancer cells in the LE. The width of connecting bands represents the strength of ligand-receptor interaction. **i** Comparative boxplot representing the number of noncancer spots directly adjacent to TC and LE spots. Non-cancer spot identity was determined based on the most enriched non-cancer cell type deconvolution. Groups were compared using a two sided Wilcoxon rank sum test with a Benjamini−Hochberg FDR correction *$p < 0.05$, **$p < 0.01$, ***$p < 0.001$, ****$p < 0.0001$. *P*-values for Cytotoxic CD8 T cells = 0.003, ecm.myCAF = 2.2e−04, intermediate fibroblast = 0.002, macrophage = 0.008 (*n* = 12 samples across 10 independent patients). Box spans 25th−75th percentiles, center line indicates median, whiskers extend to minima and maxima within 1.5*IQR. Source data are provided as a Source Data file where relevant. Abbreviations: CSC cancer stem cell, TC tumor core, LE leading edge, MET mesenchymal-epithelial transition, EMT epithelial-mesenchymal transition, CD24 (−) inverse CD24 gene expression (1/CD24), Tregs T regulatory cells.

## TC and LE gene signatures are conserved pan-cancer and are distinct in their prognostic impact

Since our annotated TC and LE regions were highly conserved across each of our OSCC samples, we wondered if the distinct molecular programs associated with these spatial regions were also present across other cancer types. We trained three machine-learning (ML) probability based prediction models on TC spots, LE spots, and all other remaining spots to generate a spatio-regional predictive model (Fig. 4a). We then applied our predictive ML model to 30 publicly available ST samples across 17 different cancer types to characterize each spot as "TC", "LE", "transitory", or "other remaining spots" in each sample (Fig. 4b). Model 10-fold cross validation revealed robust performance in all models (ROC: "TC" = 0.991, "LE" = 0.922, "transitory" = 0.943, "other remaining spots" = 0.958) with the LE region having the lowest ROC attributed to its relatively lower sensitivity of 0.694. (Fig. 4c and Supplementary Table 2). Our classifier performed well in spatially segregating cancer cell states in melanoma (SKCM), colorectal adenocarcinoma (COAD), cutaneous Squamous Cell Carcinoma (cSCC) samples, and cervical squamous cell carcinoma (CESC) samples (Fig. 4d–h). Furthermore, our classifier produced relatively consistent annotations across four serial cSCC tissue sections collected by Abalo et al.[59]. (Fig. 4d and Supplementary Fig. 4a), reinforcing the reproducibility and confidence in our classifier results. We identified highly spatially segregated LE spots in all 30 samples (Fig. 4d–h and Supplementary Fig. 4a). Pediatric medulloblastoma and hepatocellular carcinoma sections displayed the lowest proportion (1%) of inferred leading edge spots, which may indicate that these cancer types are considerably distinct from OSCC (Fig. 4d). Meanwhile, our ML algorithm trained on OSCC samples, identified spatially segregated TC spots in 15/30 publicly-available ST-profiled sections (Fig, 4d and Supplementary Fig. 4a). Our model performed particularly well in identifying TC spots in cSCC, melanoma, CESC, and COAD tissue sections, which may be attributed to the presence of keratinizing programs within these cancers (Fig. 4d, g and Supplementary Fig. 4a). These seminal results suggest that LE-associated expression states are conserved across multiple cancer contexts, while expression profiles associated with the TC are more tissue-specific.

To examine the prognostic significance of our LE and TC signatures, we incorporated samples from a bulk transcriptomic dataset (TCGA) containing matched survival data. A total of 275 HPV-negative OSCC patients were selected for survival analysis and each sample was assigned an enrichment score based on the expression of genes differentially expressed in the TC and LE (see methods; Fig. 5a). Higher TC single-sample gene-set scores were significantly associated with lower pathological stage ($p < 0.05$) (Supplementary Fig. 5a), while LE scores did not significantly vary with pathological stage ($p > 0.05$) (Supplementary Fig. 5b). Kaplan−Meier curves were generated to visualize the overall survival (OS), disease specific survival (DSS), and progression free interval (PFI) differences in relation to high or low expression of

our TC and LE signatures (Fig. 5b). High expression of the LE signature was associated with worse DSS (HR 0.60 [0.38−0.96 95% CI]; $p < 0.05$) and PFI (HR 0.67 [0.45−0.98 95% CI]; $p < 0.05$) in OSCC patients, but not OS (HR 0.81 [0.56−1.16 95% CI] $p > 0.05$) (Fig. 5b). Conversely, high expression of the TC signature was associated with improved OS (HR 1.51 [1.01−2.25 95% CI]; $p < 0.05$), DSS (HR 1.93 [1.09−3.41 95% CI]; $p < 0.05$), and PFI (HR 1.82 [1.17−2.86 95% CI]; $p < 0.05$) (Fig. 5b). To validate our results, we performed the same comparison of high and low expression of TC and LE signatures in an independent cohort of 93 HPV-negative OSCC patients (GSE41613). We found the similar trends across OS and DSS with our TC signature, but found that high expression of the LE signature was associated with both worsened OS and DSS ($p < 0.05$) (Supplementary Fig. 5c). TC and LE signatures were also very weakly negatively correlated to one another ($r = −0.17$, $p < 0.05$) (Supplementary Fig. 5d). Therefore, our TC and LE signatures displayed starkly opposed survival outcomes likely due to distinct mechanisms present in each respective niche.

As LE states appeared to be generalizable across different cancer types (Fig. 4d), we further determined the prognostic impact of our LE signature and extended our analysis to 20 common solid tumors in TCGA. We found that a high LE score was consistently associated with worse OS and DSS across multiple cancers, with the exception of breast cancer (BRCA) in OS and lung squamous cell carcinoma (LUSC) in DSS (Fig. 5c, d). Similar patterns of association between LE and PFI were seen, with the exception of melanoma (SKCM) and LUSC (Supplementary Fig. 5e).

We next explored associations between our LE and TC signatures with relevant clinical covariates to identify possible contributing factors to OSCC prognostic differences. Lower TC signature scores were associated with higher nodal stage (*p*-adj < 0.05), presence of lymphovascular invasion (*p*-adj < 0.01), higher tumor grade (*p*-adj < 0.001), positive margins (*p*-adj < 0.05), and presence of extracapsular spread (*p*-adj < 0.01); while higher LE signature scores were not associated with any clinical characteristics (*p*-adj > 0.05) (Fig. 5e, f). A weak negative correlation was also observed between EPIC CAF and our LE signature enrichment scores ($r = −0.23$, $p < 0.05$) (Supplementary Fig. 5f), which suggests that there are other unexplored mechanisms beyond CAF activity driving survival outcomes associated with the LE.

## Differentiation trajectories revealed by analyzing RNA splicing dynamics between the TC and LE

RNA velocity can be used to predict the short-term future state of individual cells using the ratio of spliced and unspliced mRNA counts[60]. Aggregating results together helps reveal the developmental trajectories of cancer cells, and similarly identify putative driver genes responsible for the transition[61]. We utilized scVelo, which builds upon RNA velocity by accounting for gene-specific transcriptional dynamics, to characterize the developmental trajectories of cancer cells present in the TC and LE. Among spatially deconvoluted cancer cells aggregated

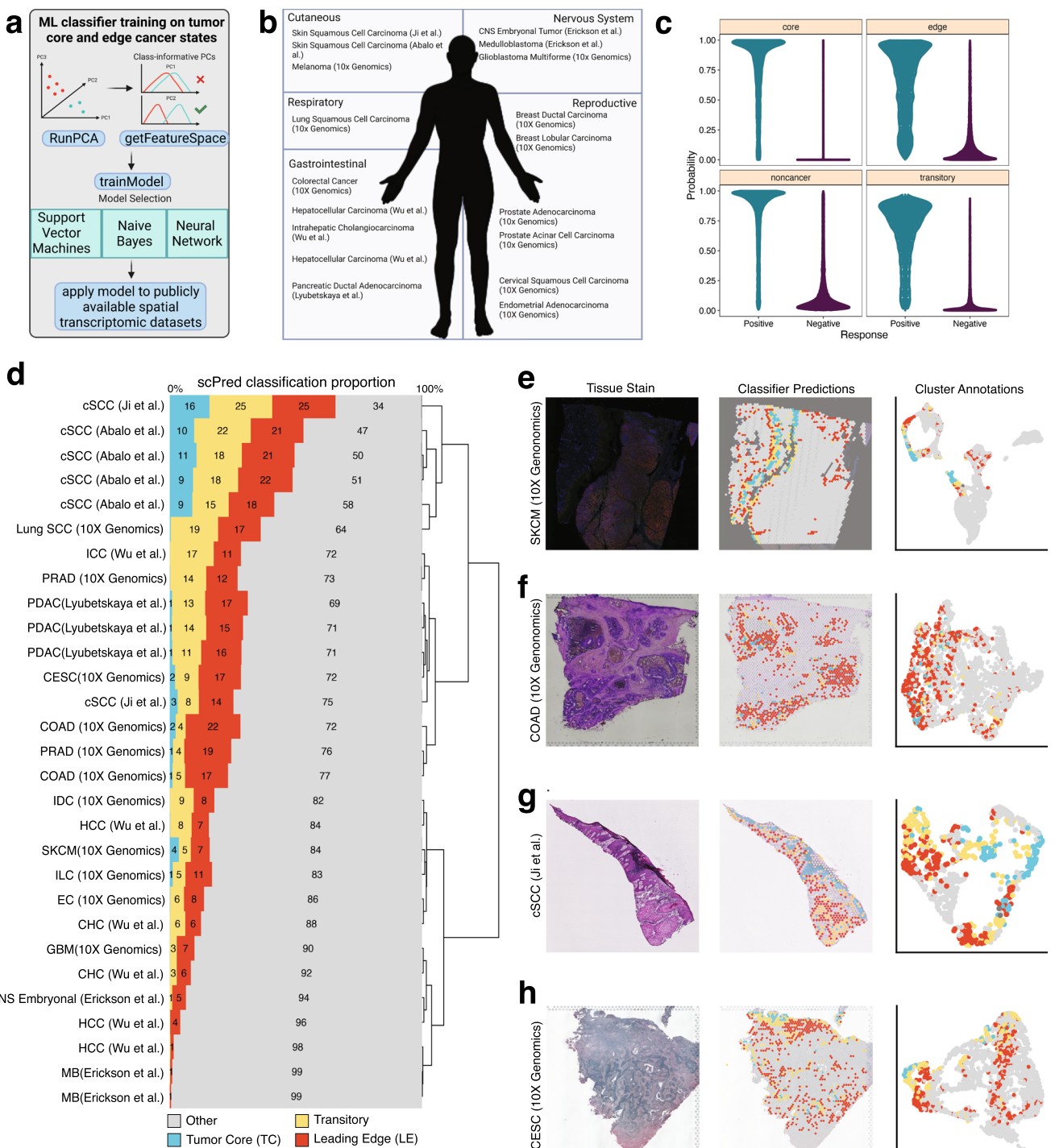

**Fig. 4 | A Machine Learning model identifies conserved TC and LE signatures across multiple cancer types. a** Infographic describing the ML strategy used for the identification of TC and LE gene signatures in publicly-available spatially-profiled samples. **b** Infographic describing publicly-available spatially-profiled samples[59,101–104,110] included in subsequent ML-training dataset. Created with BioRender. **c** Probability distribution plotted across TC, LE, transitory, and other regions. **d** Bar plot displaying the scPred classification score of each spatially distinct region across 30 different spatially profiled samples. The plot is clustered based on the similarity in predicted proportion of TC, LE, transitory, and other regions. **e**–**h** H&E-stained tissue section (left), scPred projections on stained tissue

(middle), and a UMAP colored by scPred classification (right) for cSCC, COAD, and CESC representative spatial transcriptomics testing datasets. UMAP uniform manifold approximation projection, cSCC cutaneous squamous cell carcinoma, SCC squamous cell carcinoma, ICC intrahepatic cholangiocarcinoma, PRAD prostate adenocarcinoma, PDAC pancreatic ductal carcinoma, CESC cervical squamous cell carcinoma, COAD colon adenocarcinoma, IDC invasive ductal carcinoma, HCC hepatocellular carcinoma, SKCM skin cutaneous melanoma, ILC invasive lobular carcinoma, EC endometrial adenocarcinoma, CHC combined hepatocellular and cholangiocarcinoma, GBM glioblastoma multiforme, CNS embryonal central nervous system embryonal tumor, MB medulloblastoma.

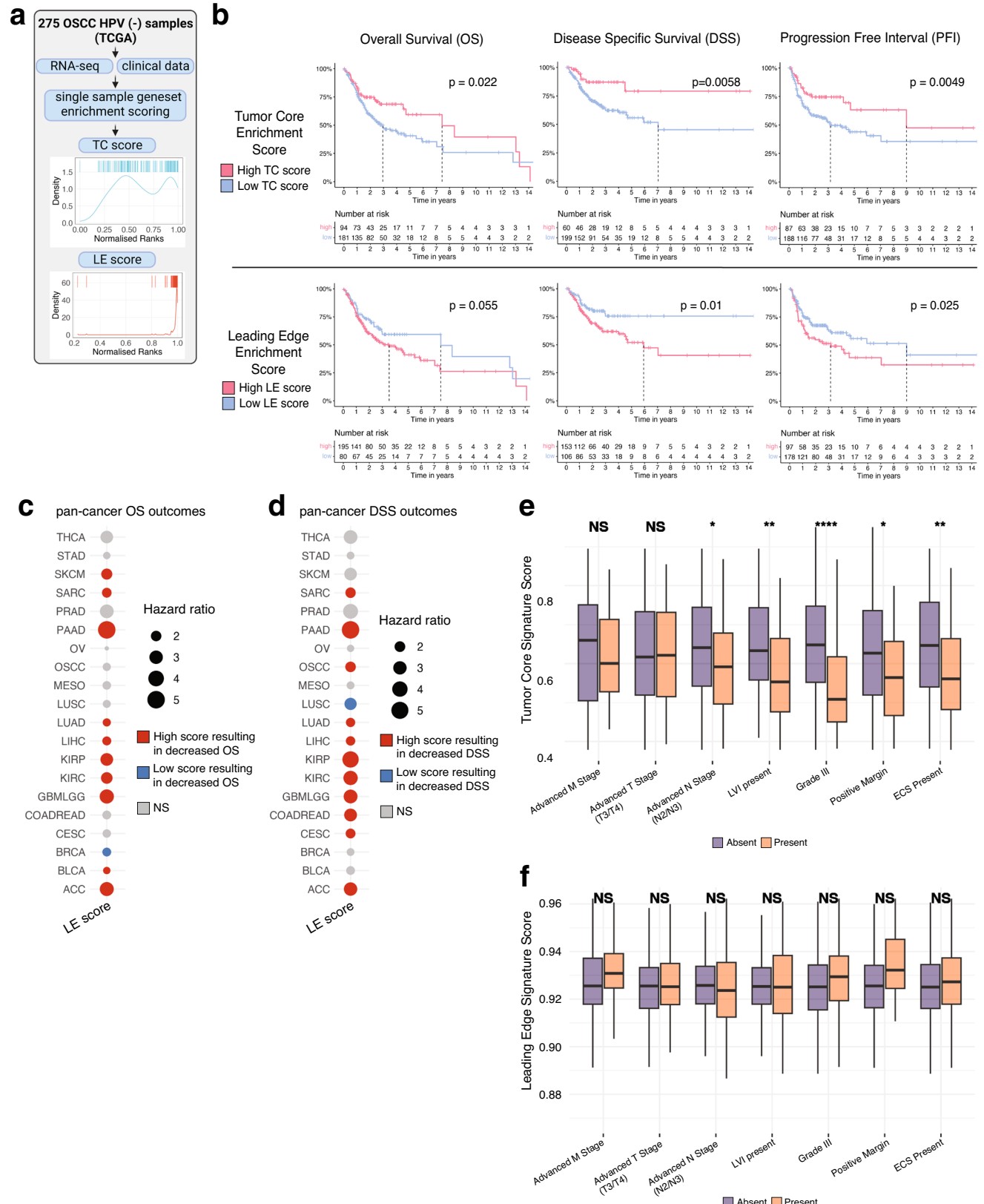

across all samples, we observed a differentiation hierarchy originating from TC extending towards LE (Fig. 6a). This hierarchy was highly reproducible and displayed high levels of agreement across spots, reflected by high spot velocity vector field confidence of greater than 0.85 in all spots (Fig. 6a). Similar patterns of directional flow were also observed at the individual patient level (Fig. 6b). Several genes displayed dynamic splicing behavior that drove the TC to LE cancer cell state differentiation (Fig. 6c). Top putative TC and LE state driver genes included *CSTA* and *IGHG3* genes, respectively (Fig. 6c, d, Supplementary Fig. 6a, and Supplementary Data 6). *CSTA* has previously been identified as a tumor suppressor gene involved in regulating mesenchymal-epithelial-transition (MET)[62]. *IGHG3* is an

**Fig. 5 | Survival associations and prognostic characteristics of the TC and LE signature. a** Infographic describing TC and LE single-sample gene-set scoring strategy for TCGA transcriptomic data. **b** Kaplan–Meier visualizations of OS, DSS, and PFI end-points stratified by TC (upper panels) and LE (lower panels) gene set enrichment scores among 275 OSCC samples. *P*-values displayed were calculated using a cox proportional hazards regression. **c, d** OS and **i** DSS pan-cancer outcomes for TC and LE gene-set enrichment scores derived from 20 common cancer types from TCGA. *P*-values and hazard ratios displayed were calculated using a cox proportional-hazard regression. **e, f** Bar plots showing the relative TC and LE enrichment score in relation to relevant clinico-pathological covariates. Significance was determined using a two-sided Wilcoxon rank sum test with a Benjamini–Hochberg FDR correction applied. Error bars represent standard error of mean (SEM). *$p < 0.05$, **$p < 0.01$, ***$p < 0.001$, ****$p < 0.0001$. *P*-values for advanced N stage = 0.018, LVI = 0.002, grade III = 1.8e−06, positive margin = 0.02,

ECS = 0.002 ($n = 275$ biologically independent HPV negative OSCC samples). Box spans 25th–75th percentiles, center line indicates median, whiskers extend to minima and maxima within 1.5*IQR. Source data are provided as a Source Data file where relevant. Abbreviations: TC tumor core, LE leading edge, THCA thyroid carcinoma, STAD stomach adenocarcinoma, SKCM skin cutaneous melanoma, SARC sarcoma, PRAD prostate adenocarcinoma, PAAD pancreatic adenocarcinoma, OV ovarian serous cystadenocarcinoma, OSCC oral squamous cell carcinoma, MESO mesothelioma, LUSC lung squamous cell carcinoma, LUAD lung adenocarcinoma, LIHC liver hepatocellular carcinoma, KIRP kidney renal papillary cell carcinoma, KIRC kidney renal clear cell carcinoma, GBMLGG brain lower grade glioma and glioblastoma multiforme, COADREAD colon adenocarcinoma/rectal adenocarcinoma, CESC cervical squamous cell carcinoma, BRCA breast invasive carcinoma, BLCA bladder urothelial carcinoma, ACC adrenocortical carcinoma, LVI lymphovascular invasion, ECS extracapsular spread.

immunoglobulin (Ig) gene that has been detected in several epithelial cancers[63,64]. While the exact mechanisms remain unclear, tumor-derived Igs have been implicated in tumor proliferation, invasion and metastasis, immune escape, and mediation of EMT-like phenotypes. These findings predict that the TC undergoes increased differentiation retaining an epithelial-like phenotype, while the LE becomes increasingly mesenchymal, and corroborates our earlier identification of CSC states. Other differentially spliced driver genes in the TC and LE included proto-oncogenes and tumor suppressor genes (Fig. 6c, and Supplementary Fig. 6a and Supplementary Data 6).

### Deriving therapeutic targets for OSCC

OSCC is characterized by its high disease burden, mortality, and treatment-associated morbidity despite improvements in diagnosis and therapeutic modalities[65]. Existing treatments such as surgery, chemotherapy and radiotherapy are effective only in a minority of patients and OSCC recurrence remains a leading cause of death[65]. Furthermore, current treatments may induce significant morbidity and reduction in the quality-of-life (QoL) related to non-specific cell death, including local defects, speech and swallowing dysfunction, and other toxic side-effects[65,66]. Although targeted therapies have been able to mitigate toxicity in other cancers, few have been employed in OSCC[65,67]. To address the dearth of effective targeted therapies in OSCC, we applied an in-silico perturbation approach to identify conserved patterns of RNA velocity that may predict therapeutic success in OSCC (Fig. 6e, f). Dynamo is an in-silico technique that is capable of accurately predicting cell-fate transition following genetic perturbation based on learned splicing vector fields[68].

We analyzed PharmacoDB drug-response data[69] for 417 drugs across at least 25 HPV negative HNSCC cell lines and found 140 drugs with drug-gene interactions identified as being upregulated or down-regulated from the DGIdb[70] (Fig. 6f and Supplementary Data 7). We then restricted our analysis to 70 drugs that demonstrated significant perturbations, and stratified drugs as "high AAC" and "low AAC" by the median (0.164; 0.026−0.560 [total range]) (Supplementary Data 7). We derived quantitative inferences for net measures of "incoming" and "outgoing" transition probabilities among tumor core and leading edge cells (Fig. 6e). Among effective drugs (high AAC), dynamo based in-silico perturbations generally displayed transition hierarchies with a reversal of the baseline state, represented by an increase in outgoing transition probabilities from the LE (Fig. 6g). This was also reflected through a significant increase in the quantitative measure of net outgoing LE transition probabilities in high AAC drugs relative to low AAC drugs ($p < 0.05$) (Fig. 6h). These findings were also replicated on an individual drug level (Fig. 6i, j). An inverse phenotype was seen among several ineffective drugs (low AAC), which displayed similar transition hierarchies to the baseline state (Fig. 6k, l). Interestingly, dynamo based in-silico perturbations of common immunotherapy targets (anti-PD-1, anti-CTLA-4) displayed similar results to effective drugs (high AAC), with a predominance in outgoing LE transition signaling

(Supplementary Fig. 6b, c). Although trending, no statistically significant difference was observed across incoming transition signals at the TC ($p > 0.05$) (Supplementary Fig. 6d); several outliers were present among effective drugs which drive a lack of significance. Therefore, effective drugs may generally manifest their efficacy by inducing reversal from the LE state, with some drugs specifically advancing the transformation into the TC state. These findings were further confirmed following drug class stratification, which found significant differences in LE outgoing and TC incoming transition probabilities (Supplementary Fig. 6e, f). However, several drug classes were underpowered to derive conclusive biological inferences (Supplementary Fig. 6e, f and Supplementary Data 7).

### Discussion

Intratumoral heterogeneity is a leading determinant for treatment failure and poor survival outcomes in cancer patients[71,72]. However the spatial underpinnings of heterogeneity in the tumor microenvironment are poorly understood. Although it is appreciated that the TC and LE profoundly affect tumor biology, the gene expression profiles of these compartments and their effects on creating a heterogeneous tumor microenvironment have not been systematically explored. Here, we leverage spatial transcriptomics profiling of 12 OSCC tissue samples to extensively characterize the TC and LE transcriptomes to better unravel their contribution to OSCC development, progression and invasion.

Firstly, we find that malignant OSCC TC and LE spots represent distinct spatial architectures with unique functional characteristics that are conserved across patients. The TC is characterized by a keratinized and differentiated state, while the LE confers several invasive and metastatic properties. These spatial architectures are also valuable for prognostication–high LE signature scores and low TC scores were associated with worse patient outcomes, while high TC scores exert a protective effect that may be partially attributed to increased tumor differentiation. Interestingly, these spatial architectures are generalizable across cancer types. Through our ML model, we find that the LE is conserved in multiple different cancers and can be annotated with considerable accuracy, while the programs present in the TC may be restricted to tissues with similar origins. Furthermore, a similar trend in the prognostic association of our LE signature and TC signature was observed: the LE signature is prognostic across many cancers, while the TC signature is less specific and prognostic across fewer cancer types. Overall, we have identified a pan-cancer conserved LE-associated transcriptional program that is associated with worse patient outcomes.

We also report that the differences in gene expression between the TC and LE are not governed by HNSCC molecular subtype compositions, or genetic differences from different tumor evolutionary clonal lineages; our results suggest that multiple HNSCC subtypes may be represented within a single tumor sample. We propose that the transcriptomic differences in the TC and LE are driven by the existence of spatially unique cancer cell states. Through RNA velocity analysis,

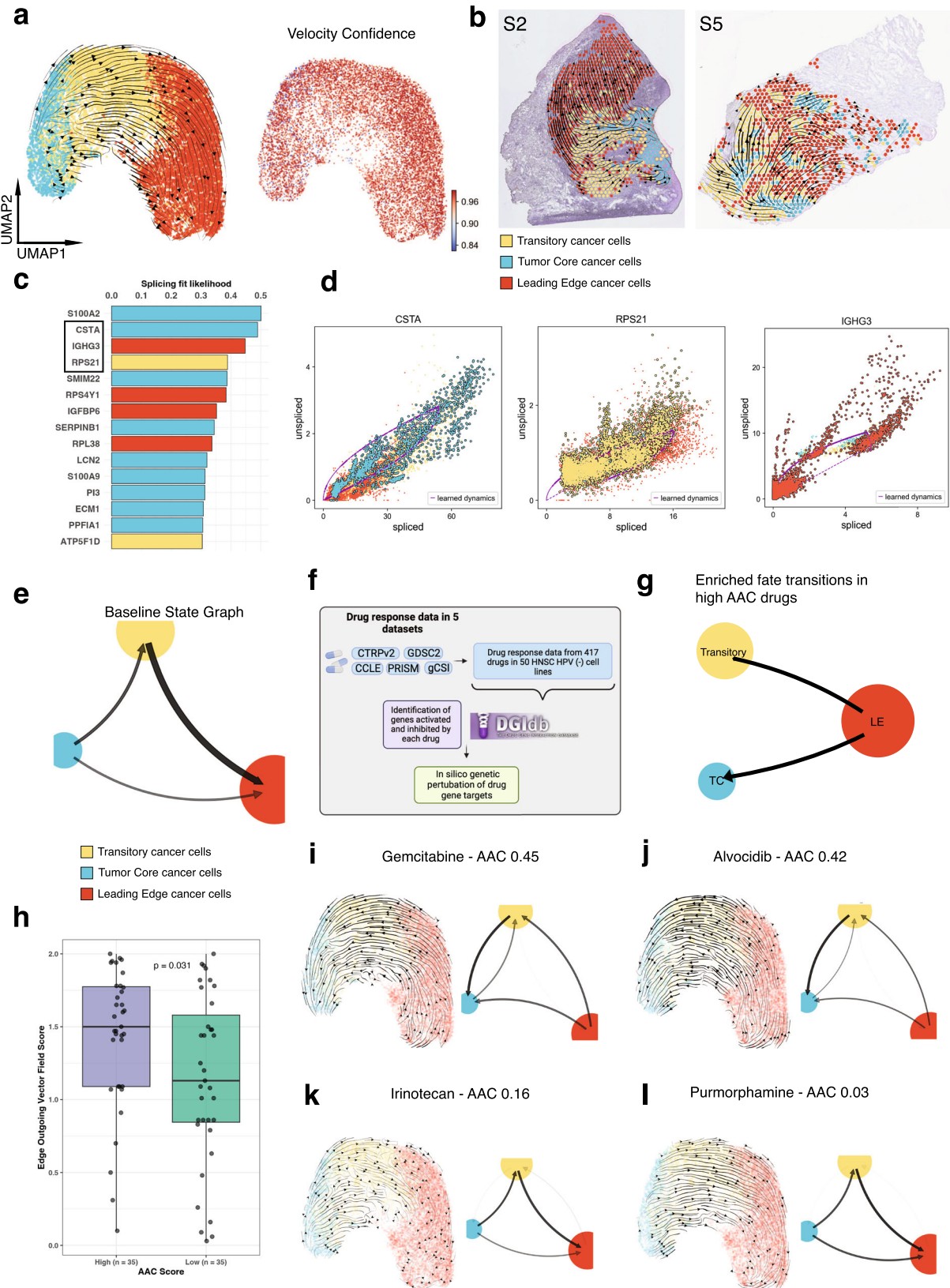

we find that these spatially unique cancer states are governed by a differentiation hierarchy composed of progenitor-like cancer cell states from the TC developing into more specialized cancer cell states of the LE. Interestingly, we also observed similar trends analyzing the cellular states of the CSC contexture in the TC and LE: mesenchymal-like CSCs populate the LE, while epithelial-like CSCs reside in the TC.

Taken together, we believe that cancer cells from the TC state can transition into LE state by gradually acquiring a more aggressive EMT-like phenotype that promotes cancer invasion and dissemination. Although multiple previous studies have described the presence of transcriptionally unique cancer cell states[13,53,73,74], our research shows that these states are also spatially regulated.

**Fig. 6 | Analysis of RNA splicing dynamics reveals differential developmental trajectories and therapeutic vulnerabilities in the TC and LE. a** UMAP of spatially deconvolved cancer cell spots, with overlaid RNA velocity streams, colored based on TC, transitory, and LE annotations, and UMAP plot with overlaid RNA velocity confidence. **b** Representative spatially profiled samples (samples 2 and 5) overlayed with RNA velocity streams and colored by TC and LE cancer cell annotations. **c** Bar plot visualizing top differentially spliced genes (genes with dynamic splicing behavior) within TC and LE regions. Bars are colored by whether a higher proportion of the gene exists in its spliced form in the TC or LE. **d** Phase portraits showing the ratio of spliced and unspliced RNA for top differentially spliced genes, purple lines depict predicted splicing steady state. **e** Cell fate transition probability state graph for TC, transitory, and LE annotations as inferred by vector field integration. **f** Infographic describing strategy used to systematically collect drug response data and test in silico drug perturbations. Created with BioRender. **g** Infographic state graph plot highlighting cell fate transitions enriched in high AAC drugs. **h** Boxplot comparing edge outgoing vector field strengths between high AAC and low AAC drugs stratified based on median. AAC groupings are compared using a two-sided Wilcoxon rank sum test ($n = 70$ independent drugs). Box spans 25th–75th percentiles, center line indicates median, whiskers extend to minima and maxima within 1.5*IQR. UMAPs showing the resultant vector field and state graphs following in silico perturbations of two targets of (**i, j**) high (effective) AAC anticancer drugs, and (**k, l**) low AAC anticancer drugs. Source data are provided as a Source Data file where relevant. Abbreviations: AAC area above curve.

Finally, we illustrate an application of RNA velocity to ST data to predict drug response. We find that effective drugs (high AAC values) were generally enriched for outgoing transition probabilities responsible for inducing a reversal in the LE state, highlighting the prominence of the LE as a global therapeutic target. Several drugs may further confer their efficacy by inducing cancer cell transformation into the TC state. We hypothesize that effective anticancer drugs direct the transition from a LE-like cancer cell state to a TC-like cancer cell state, therefore making these cells less phenotypically less aggressive by suppressing invasive/metastatic signaling. Alvocidib is a CDK inhibitor currently under investigation for its use in acute myeloid lymphoma (AML)[75], which demonstrated above average outgoing LE and incoming TC transitory signals in our analysis. As CDK inhibitors have been previously associated with promising results in OSCC cell lines[76,77], Alvocidib may be a promising candidate for further research[78,79]. Importantly, the results of our perturbation analysis illustrate how spatial transcriptional profiles may be leveraged to assess drug efficacy in-silico and demonstrate that the tumor LE is a credible drug target.

Our work seeks to characterize and uncover unique spatial patterns of gene expression between the TC and LE, and represents an application of ST to OSCC. Several exciting avenues of study remain, including opportunities to better understand the conserved features in cancer invasion and metastasis, and guide subsequent treatment efforts targeted at the LE with a pan-cancer scope. We believe that the spatiotemporal mechanistic insights gained from our study will help direct the development of improved targeted therapies for OSCC and beyond.

## Methods
### Sample collection and annotation
Fresh-frozen OSCC tissue was obtained from the Alberta Cancer Research Biobank under the approval of the Health Research Ethics Board of Alberta – Cancer Committee (Study ID: HREBA.CC-16-0644). Informed consent for tissue collection and research was obtained from each patient. No participant compensation was provided. Tumor samples were collected at the time of the surgery, embedded in optimal cutting temperature (OCT) compound, and stored frozen at −80 °C until retrieval for the study. The study samples were sectioned on a cryostat into 10 μm sections, stained with hematoxylin and eosin (H&E) and used for ST. The H&E sections were used to annotate the tissues by the study pathologist (M.H.). Regions were labeled as: squamous cell carcinoma (SCC), lymphocyte-positive stroma, lymphocyte-negative stroma, normal mucosa, glandular stroma, muscle, keratin, artery/vein, and artifact. Pathologist annotations were exported from the Loupe bBrowser (Version 6) and imported into Seurat for further analysis.

### Spatial transcriptomic profiling
Tissue optimization was performed to determine the optimal permeabilization time for OSCC tissue for the downstream gene expression protocol. Spatial transcriptomics was performed on OSCC cryosections using the Visium Spatial Gene Expression Slide & Reagent Kit, 16 reactions (Catalog # PN-1000184), according to the manufacturer's protocol (10x Genomics, Pleasanton, CA, USA). Briefly, OCT-embedded 10 micrometer-thick cryosections of OSCC samples were placed on the Visium spatial slide. Sections were enzymatically permeabilized for 24 min. cDNA was obtained from mRNA bound to capture oligos printed on the slide. cDNA quantification was performed using Agilent Bioanalyzer High Sensitivity Kit (Catalog # 5067-4626) on an Agilent Bioanalyzer 2100 (Agilent Technologies, CA, USA). cDNA libraries were sequenced on an Illumina NovaSeq 6000 sequencer using the SP flowcell (200 cycles) at the Centre for Health Genomics and Informatics (CHGI, University of Calgary, Alberta, Canada). Sequencing reads were aligned using the 10x Genomics Space Ranger 1.3.1 pipeline to the standard GRCh38 reference genome. 12 samples passing alignment QC were aggregated together using the 10x Genomics Space Ranger aggr function to normalize for read depth between samples. Aggregated samples ($n = 12$) recovered a total number of 24,876 spots containing tissue sequenced to 43,648 post-normalization mean reads per spot.

### Spatial transcriptomic data pre-processing
The aggregated HDF5 matrix was imported into R and split by sample. Feature-barcode matrices for each sample were imported into the R package 'Seurat' (Version 4.2.0) for normalization, quality control, batch effect correction, dimensionality reduction, and Louvain clustering[80]. Spots expressing less than 200 features were excluded from downstream analysis. Sample level normalization was performed using the SCTransform function in 'Seurat' (Version 4.3.0). Batch effect correction, integration, and dimensionality reduction was performed using the introduction to scRNA-seq integration Seurat vignette (Version 4.3.0) (https://satijalab.org/seurat/articles/integration_introduction.html) with no deviations.

### Celltype deconvolution and malignancy annotation
Single-cell HNSCC data was downloaded from GSE103322 and imported into R using the 'CreateSeuratObject' function in 'Seurat' (Version 4.3.0). In brief, the single-cell data was normalized using 'SCTransform', dimensionality reduced, and Louvain clustered. Fibroblast subtypes were assigned to Louvain clusters; *ACTA2+TAGLN+* cells were annotated as Myofibroblasts and *CXCL1+PDPN1+ACTA2- MMP1-* cells were annotated as Intermediate Fibroblasts based on marker genes described in the datasets original paper[13]. CAF subtypes conserved in HNSCC were annotated using the marker genes *LRRC15* and *GJB2* for ecm-MYCAFs and *ADH1B* and *GPX3* for detox-iCAFs[16]. The 'CARD' R package (Version 1.0)[81] was used to perform deconvolution of spatial transcriptomic spots into cell types with the 'CARD_deconvolution' function. The GSE103322 single cell dataset was used as reference while the spatial transcriptomics data was submitted as a query dataset. Celltype deconvolution data was added to Seurat object metadata for downstream analysis.

The 'numbat' R package (Version 1.1.0)[82] was used to conduct haplotype-aware CNV inference on all 12 spatial transcriptomics

objects from raw ST BAM files. CNV inference was conducted in accordance with the numbat vignette for spatial transcriptomics data with no deviations (https://kharchenkolab.github.io/numbat). Mutant versus normal probability (p_cnv) was exported from numbat and added to 'Seurat' metadata for downstream analysis.

Spots were annotated as "cancer cells" if they had a 'CARD' deconvolution proportion of greater than 0.99, or a numbat CNV based mutant versus normal probability (p_cnv) of greater than 0.99. Spots annotated as 'cancer cells' also must have been annotated as containing squamous cell carcinoma by the study pathologist. Spots not annotated as 'cancer cells' were annotated for cell type based on the largest non-cancer deconvolution proportion as inferred by CARD.

## TC, transitory, and LE annotation
To investigate cancer cell heterogeneity, we reanalyzed only cancer cells from all ST slides, using 'Seurat' (Version 4.3.0) for normalization, batch effect correction, dimensionality reduction, and Louvain clustering with a resolution of 1.0. To identify hierarchical states of cancer cell identity, a phylogenetic tree was constructed in 'Seurat' (Version 4.3.0) using the 'BuildClusterTree' function. This tree was visualized using 'plot.phylo' function within the ape R package (Version 5.6-2)[83], revealing three hierarchical divisions of ST cancer cells, which were subsequently annotated and visualized within 'Seurat' (Version 4.3.0). Broad differential gene expression analysis between the three nodal clusters was conducted using the 'FindAllMarkers' Seurat (Version 4.3.0) function which implements a two-sided Wilcoxon Rank Sum test with a Bonferroni correction. Visualization of differentially expressed genes was performed using the 'SCpubr' R package (Version 1.1.2.9000)[84]. Nodal clusters were annotated as 'edge', 'core' and 'transitory' based on differentially expressed genes and the localized expression of the leading edge markers *LAMC2* and *ITGA5* and tumor core markers *CLDN4* and *SPRR1B* as previously validated in head and neck cancer[13]. Kernel density diagrams for marker genes were constructed using the 'do_NebulosaPlot' in the 'SCpubr' R package (Version 1.1.2.9000)[84], employing the R package 'Nebulosa' (Version 1.8.0)[85].

## Differential expression analysis, consensus plotting, and correlation heatmaps
Differentially expressed genes were identified using a two-sided Wilcoxon rank-sum test with a Bonferroni correction in Seurat (Version 4.3.0) with a logFC > 0.25 and adjusted $P$ value < 0.001. Stacked bar plots showing cumulative expression log2FC for each gene across all samples were generated using an adaptation of the constructConsensus function[86]. Consensus plots were created to display the top 25 genes differentially expressed in more than 9 samples (>9/12 samples). Differentially expressed genes for each sample were imported into Ingenuity Pathway Analysis (IPA) for pathway enrichment analysis. IPA exports were imported into the multienrichjam R package (Version 0.0.57.9) (https://github.com/jmw86069/multienrichjam). The multienrichjam mem_enrichment_heatmap function was modified to create pathway enrichment plots across samples if pathways were activated or deactivated across 10 or more samples. Whole transcriptome average expression was Pearson correlated using the 'cor' function in the R 'stats' package (version 4.2.2). Correlation values were plotted using the Heatmap function in the 'ComplexHeatmap' R package (Version 2.14.0)[87].

## Statistical approach for comparing scores across sample groups
Differentially regulated hallmark pathways between TC and LE cancer cell states were identified by modifying code from the 'SCPA' R package (Version 1.2.0)[88]. The 'compare_seurat' function queried hallmark genesets in 'Seurat' (Version 4.3) R objects housing integrated data and tested for differential pathway activity using multivariate distribution testing with a Bonferroni correction applied. Plots comparing hallmark

genesets were created using the 'ggplot2' (Version 3.4.0) and 'ggrepl' R packages (Version 0.9.2).

Select hallmark cancer gene signatures were scored in each spot using the Seurat (Version 4.3.0) function 'addmodulescore'. To test for differences in the module scores calculated between the TC and LE, a two-sided paired Wilcoxon rank sum test was conducted using the 'ggpubr' (Version 0.5.0) function 'stat_compare_means' and corrected using a Bonferroni correction. To perform the scoring of broad CSC, epithelial CSC and mesenchymal CSC state signatures, an expression module was programmed using the Seurat (Version 4.3.0) function 'addmodulescore'. CSC gene characteristics were identified from literature according to their predicted upregulation or downregulation[55]. The scoring contribution of genes predicted to be upregulated and downregulated were then cumulatively added to derive module scores. Kernel density diagrams for module scores were constructed using the 'do_NebulosaPlot' in the 'SCpubr' R package (Version 1.1.2.9000), employing the R package 'Nebulosa' (Version 1.8.0).

## Gene regulatory network inference
The Python implementation of the SCENIC[89] pipeline (pySCENIC version 0.12.1)[90] was used to infer regulatory interactions between transcription factors and their targetomes. Core and edge cancer cell data was processed using default and recommended parameters specified in pySCENIC's tutorial (https://pyscenic.readthedocs.io/en/latest/index.html) and regulons were pruned using the hg19 RcisTarget database. pySCENIC outputted regulon activity scores were added to a 'Seurat' (version 4.3) object using the 'CreateAssayObject' function. A stacked bar plot showing the cumulative logFC of AUCell scores for each TF across all samples was generated using an adaptation of the constructConsensus function with an adjusted $p$ value < 0.05.

## Immunofluorescence and fluorescence cell-based analyses
The OCT-frozen tissue was sectioned at 6μm thickness and placed on a glass slide. The sections were air-dried at room temperature for 30 min followed by fixation using 4% paraformaldehyde for 15 minutes. After a brief wash with 1X PBS, the sections were blocked using 10% Horse serum for 1 h. An antibody dilution buffer composed of PBS, 1% BSA, 0.1% cold fish skin gelatin, and 0.1% Triton X-100 was prepared and the sections were subjected to indirect immunofluorescence staining using a rabbit polyclonal anti-CD24 antibody (Abcam: 1:250 dilution; Catalog # ab244478) and mouse monoclonal anti-CD44 antibody (IM7, Invitrogen: 1:250 dilution; Catalog # 14-0441-82) as the primary antibodies. Knockout validation was provided by the manufacturer[91,92], and dilution specifications were chosen according to manufacturer protocols. Afterwards, we included corresponding fluorophore-conjugated secondary antibodies goat anti-rabbit antibody conjugated to Alexa 546 and donkey anti-rat antibody conjugated to Alexa 488 (1:500; Jackson ImmunoResearch Laboratories) alongside Hoechst 33342 (1:1000). The sections were also incubated with the DNA fluorescent dye Hoechst 33342 (Invitrogen; Catalog # H3570) to visualize cell nuclei. To attach coverslips to the slides, we used Fluoromount-G (SouthernBiotech; Catalog # 0100–01), and fluorescence images of the tissue sections were captured using a fluorescence microscope at 10X magnification (Zeiss AxioObserver Z1). Exposure times for CD24 (2 s), CD44 (200 ms), and Hoechst-specific (6 ms) signals were kept constant across samples. A serial section for each sample was stained with hematoxylin and eosin (H&E) stains and examined by the study pathologist (M.H.) with a brightfield microscope at 10X magnification (Olympus IX70) to locate the TC and LE/stroma.

## TCGA analysis and gene-set scoring
Patient metadata, survival data, and bulk RNA sequencing data was downloaded for all samples in The National Cancer Institute's Cancer Genome Atlas using the 'UCSCXenaTools' R package (version 1.4.8)[93,94]. HPV negative OSCC samples from 275 patients were identified based

on patient metadata. Gene set enrichment analysis was performed on genes differentially expressed in the TC and LE in at least 9/12 patients having an adjusted *p* value < 0.05 using the singscore R package with default parameters[95]. The optimal quantile cut point for core and edge geneset scores was determined using the 'surv_cutpoint' function in the 'survminer' R package (Version 0.4.9) with a minimum proportion of 0.2. Kaplan–meier survival plots were generated using the 'survfit' function in the 'survival' R package (Version 3.4-0) and plotted using the 'ggsurvplot' function in the 'survminer' R package (Version 0.4.9) (https://github.com/kassambara/survminer). Hazard ratios and *p*-values were obtained from cox proportional hazard testing using the 'coxph' function in the 'survival' R package (Version 3.4-0).

When comparing survival across multiple cancer types, the optimal quantile cut point for the LE gene set score in each cancer was determined using the 'surv_cutpoint' function in the 'survminer' R package (Version 0.4.9) with a minimum proportion of 0.1. Optimized hazard ratios and *p*-values were subsequently calculated using the 'coxph' function in the survival (Version 3.4-0) R package with default parameters.

To further evaluate our TC and LE gene-sets across multiple relevant clinical features, we compared TC and LE enrichment scores across the presence/absence of multiple clinical features, testing for significance using two-sided Wilcoxon Rank sum test with a Benjamini–Hochberg correction applied with the 'stats' R package (Version 4.2.2).

To test for relationships between our TC and LE scores, and our LE score and a CAF score, we implemented a Pearson correlation test in our dataset using the 'ggpubr' (Version 0.5.0) 'stat_cor' function and plotted using the 'ggscatter' function. A CAF score was identified by using 'EPIC' based deconvolution in the 'immunedeconv' R package (Version 1.1.5) with default settings[96].

### TCGA validation
To validate our findings in an external database, data from GSE41613 containing 93 HPV negative OSCC patients was downloaded and imported into R[97]. Intensity data was preprocessed and background corrected using the 'limma' R package (Version 3.54.1)[98]; gene-sets were scored with 'singscore' (Version 1.18.0), stratified into high and low TC and LE scores using an optimized cutpoint, and had their optimized hazard ratios and *p*-values calculated.

### TCGA subtyping
TCGA subtype data for 275 OSCC samples was downloaded using the 'PanCancerAtlas_subtypes' function from the 'TCGAbiolinks' R package (Version 2.25.3)[99]. Bulk RNA sequencing data and subtype data from the 'TCGAbiolinks' package (Version 2.25.3) was added to a Seurat (Version 4.3.0) object using the 'CreateSeuratObject' function[99]. Deconvolution of spatial transcriptomic spots into HNSC subtypes was performed using the Seurat vignette for mapping and annotating query datasets with no modification (https://satijalab.org/seurat/articles/integration_mapping.html). The subtyped data was held as a reference dataset and the spatial transcriptomics data submitted as a query dataset.

TCGA subtype data was scored using the 'singscore' R package (Version 1.18.0) for TC and LE gene-sets. Scores across HNSC subtypes were compared using a Kruskal–Wallis one way analysis of variance test implemented through the 'ggpubr' (Version 0.5.0) function 'stat_compare_means'[95].

### Machine learning model for cancer cell states
To assess whether the core, transitory, and edge states we identified in OSCC were conserved in other solid tumors, we trained three machine-learning probability-based prediction models (Support Vector Machines with Radial Basis Function Kernel, Model Averaged Neural Network, and Naive Bayes) using 'scPred' (Version 1.9.2)[100].

Briefly, feature selection was performed by shortlisting top 50 class-informative PCs distinguishing spot-level variance between 'core', 'transitory', 'edge', and 'noncancer'. Prediction models were trained using the 'caret' R package (Version 6.0-93). Training probabilities for spatial tumor states in the OSCC dataset were evaluated using 'get_scpred' and visualized using 'plot_probabilities'. Support Vector Machines with Radial Basis Function Kernel was used to predict core, edge, and noncancer spots, while Naive Bayes was used to predict the transitory cell state. 2 combined hepatocellular cholangiocarcinoma (CHC) (Wu et al.[101]), 3 hepatocellular carcinoma (HCC) (Wu et al.[101]), 1 intrahepatic cholangiocarcinoma (ICC) (Wu et al.[101]), 2 cutaneous squamous cell carcinoma (cSCC) (Ji et al.[102]), 1 glioblastoma (GBM) (10X Genomics), 1 human breast cancer Invasive ductal carcinoma (HBC-IDC) (10X Genomics), 1 human breast cancer Invasive lobular carcinoma (HBC-ILC) (10X Genomics), 1 colorectal cancer (CRC) (10x genomics), and 1 ovarian cancer (OV) (10x Genomics), 1 lung squamous cell carcinoma (10X Genomics), 4 cutaneous squamous cell carcinoma (Abalo et al.[59]) 1 prostate adenocarcinoma, invasive ductal carcinoma (10x Genomics), 3 pancreatic ductal adenocarcinoma (Lyubetskaya et al.[103]) 1 cervical squamous cell carcinoma (10X Genomics), 1 prostate acinar cell carcinoma (10x Genomics), 1 intestinal colorectal cancer (10X Genomics), 1 melanoma (10X genomics), 1 pediatric CNS embryonal tumor (Erickson et al.[104]), and 2 pediatric medulloblastoma samples (Erickson et al.[104]) were classified and Harmony-integrated using 'scPredict' with default probability threshold[101,102,105]. scPred-generated annotations were imported into 'Seurat' (Version 4.3.0) and overlaid on top of histologic image using the 'SpatialDimPlot' function. The trained model used for classification is publicly available via our Figshare portal (https://doi.org/10.6084/m9.figshare.20304456.v1).

### Inferring cell communication networks and analyzing cell neighbors
The 'CellChat' R package (Version 1.6.1) was used to infer cell-cell interaction networks from a Seurat object containing deconvoluted spots using the 'createCellChat' function[106]. Filtered Circos plots were generated using the 'netVisual_chord_gene' function in 'CellChat' to visualize ligand-receptor pairs, circos plots were modified for clearer visualization by thresholding based on communication probability (core to core: 0.001, edge to edge: 0.005, CAF and edge: 0.05). Circos plots visualizing specific pathways were generated using the 'netVisual_aggregate' function. Cellchat was also used to compare the overall information flow of the core and edge in different signaling families using the 'rankNet' function in comparison mode[106].

Differences in the absolute count of non-cancer spots neighboring our annotated tumor core and leading edge cancer cells were determined by classifying neighboring spots according to the greatest non-cancerous cell type deconvolution proportion, followed by counting of the annotated spots neighboring the TC and LE. Neighboring spots were defined as spots directly in contact with our previously annotated malignant TC and LE spots. These counts were then compared in number using a two-sided Wilcoxon Rank sum test with a Benjamini–Hochberg applied using the 'rstatix' R package (Version 0.7.1) and the 'stats' R package (Version 4.2.2).

### Constructing cellular trajectories using RNA velocity
To generate spliced and unspliced assays used to infer RNA velocity, the command line interface tool in 'velocyto' (Version 0.17.16) was employed (https://velocyto.org)[107]. For each sample, barcodes and BAM files corresponding to pathologist annotated cancerous regions were supplied to the velocyto 'run' command. The velocyto 'run' command was provided with a .gtf gene annotation file that was created using the cellranger 'mkref' function applied to the standard GRCh38 reference genome. Output loom files were combined and

imported into the 'scVelo' python package (Version 0.2.5) for further analysis, dimensionality reduction coordinates and sample metadata for scvelo analyses were imported from Seurat (Version 4.3.0)[61]. Multiple scVelo objects were processed identically in parallel. Spot-level velocity was derived from the scVelo dynamical model which identifies spot-level trajectories based on the ratio of spliced mRNA to un-spliced pre-mRNAs. First and second order velocity vector moments were calculated using 'scv.pp.moments (data,n_pcs=10, n_neighbors=30)'. Dynamical velocity was calculated using the 'scv.tl.recover_dynamics' function and the 'scv.t.velocity' functions. The velocity confidence, based on velocity vectors, was computed using the 'scv.tl.velocity_confidence' function. The 'scv.tl.velocity_confidence' function determines the level of agreement between spot-level velocity vectors and their neighboring spots using a correlation-based approach. Velocity confidence results were visualized with the 'scv.pl.scatter' function. Differential splicing for each gene was calculated using 'scv.tl.rank_velocity_genes' function which utilizes a two-sided Welch t-test with overestimated variance to compare multiple grouping variables and identify group-specific rank ordered gene lists. Data from 'scVelo' was subsequently imported into the 'Dynamo' python package (Version 1.2.0)[68] for vector field learning and spot fate trajectory inference with no deviation from their standard vignette.

### Drug response and in-silico perturbation of RNA velocity trajectories

Cell line drug response data was downloaded from the CTRPv2, GDSC2, PRISM, CCLE, and gCSI databases using the R package 'PharmacoGx' (Version 3.2.0)[108]. Fifty HPV negative HNSCC cell lines were identified across all five drug datasets. 417 drugs were identified to have AAC values available in at least 25 HPV negative HNSCC cell lines. AAC values for each cell line were averaged across datasets for datasets with overlapping cell lines. The ACC value for each drug was calculated by taking the 10% trimmed mean of the ACC values observed across all cell lines to control for drug response outliers. Drug names were subsequently passed to the drug gene interaction database API (Version 2) to identify their downstream targets. Therapeutics that did not include information on upregulation or downregulation were removed from subsequent analysis. Upregulation was determined based on the DGIdb keywords: "activator", "agonist", "inducer", "partial agonist", "positive modulator", "potentiator", and "stimulator". Downregulation was determined based on the DGIdb keywords: "inhibitor", "antagonist", "partial antagonist", "blocker", "inverse agonist", "negative modulator", and "suppressor". 143 drugs with downstream targets were identified. Drugs were included in further analyses if their downstream targets had non-zero expression in our spatial transcriptomics dataset. Drug targets were passed to the python package 'Dynamo' (Version 1.2.0) for in silico perturbation analysis. Drug gene targets were perturbed in silico using the 'dyn.pd.pertrubation' with a Jv scaling factor of −200 for downregulated genes and 200 for upregulated genes. Velocity vector fields resulting from perturbation were integrated to estimate a quantitative spot fate transition probability between core, transitory, and edge states using the 'dyn.pd.state_graph' function in 'vf' mode, and plotted using the 'dyn.pl.state_graph function. The 'edge outgoing' signature was generated by adding the spot fate transition probability of 'edge' to 'transitory' and 'edge' to 'core'. The 'core incoming' signature was generated by adding the spot fate transition probability of 'edge' to 'core' and' transitory' to 'core'. Drugs were classified as having a 'high' AAC values if they had an AAC greater than the median, or a 'low' AAC values if they had an AAC less than the median. Spot fate transition probability scores were compared across 'high' and 'low' AAC groups using a two-sided paired Wilcoxon rank sum test with the 'ggpubr' (Version 0.5.0) function 'stat_compare_means'. Immune check-point genes CTLA4 and CD274 were separately perturbed in silico using 'dyn.pd.pertrubation' with a Jv scaling factor of −1000.

Drug mechanism of action was identified using the PRISM database in the R package 'PharmacoGx' (Version 3.2.0). Drug names were aligned with the PRISIM dataset through the Levenshtein distance algorithm, executed via the 'adist' function from the 'Utils' package (Version 4.2.2) in R. This approach facilitated the identification of drug names with high similarity, permitting a maximum character-level discrepancy of three, encompassing substitutions, insertions, deletions, or capitalization changes. Drugs with identified classes were defined by groups of drugs ($n > 1$) sharing a common mechanism of action. Spot fate transition probabilities were compared across drug classes using a Kruskal–Wallis one way analysis of variance test implemented through the 'ggpubr' (Version 0.5.0) function 'stat_compare_means'.

### Spatial transcriptomics atlases

A web portal was created to enable exploration of all 12 spatial samples. This portal was built using 'shiny' (Version 1.7.4), 'shinyLP' (Version 1.1.2), and 'shinythemes' (Version 1.2.0) R packages. The portal is available for public access at http://www.pboselab.ca/spatial_OSCC/. A portal for exploring in-silico perturbation approaches is available for public access at www.pboselab.ca/dynamo_OSCC. Our in-silico portal additionally used the R package 'reticulate' (version 1.2.6), and the python package 'Dynamo' (Version 1.2.0).

### Statistics & reproducibility

No statistical method was used to predetermine sample size. Two collected samples were excluded from analysis due to poor sequencing quality. The experiments were not randomized. The Investigators were not blinded to allocation during experiments and outcome assessment.

### Reporting summary

Further information on research design is available in the Nature Portfolio Reporting Summary linked to this article.

## Data availability

The Raw and SpaceRanger processed spatial transcriptomics data generated in this study have been deposited in the National Center of Biotechnology Information's Gene Expression Omnibus (GEO) database under accession code GSE208253. The processed Seurat objects, loom files generated by velocyto, and the scPred prediction model data are available at https://doi.org/10.6084/m9.figshare.20304456.v1. Spatial datasets are also available for public access at our companion portal http://www.pboselab.ca/spatial_OSCC/. In silico perturbation results are available for public access at www.pboselab.ca/dynamo_OSCC. The analyzed spatial transcriptomics differential expression, transcription factor, hallmark pathway, cellChat, differential splicing, and drug perturbation data generated in this study are provided in the Supplementary Information/Source Data file. The Hallmark gene-sets for core-edge testing data used in this study are available in the Molecular Signatures Database v7.5.1[109]. The P-EMT gene-set and single-cell HNSCC data used for deconvolution data used in this study are available in the GEO database under accession code GSE103332[13]. Cancer stem cell gene-set data used in this study were extracted from literature[52,55]. GEO: bulk RNA-sequencing data and associated clinical data used in this study are available in the National Cancer Institute's The Cancer Genome Atlas database through UCSC Xena [https://xena.ucsc.edu][93]. The validation genomic survival dataset data used in this study are available in the GEO database under accession code GSE41613[97]. The scpred analysis data used in this study are available in the TheLifeome, GEO, Mendeley, and 10X Genomics databases under accession codes Lifeome: 7:eabg3750, GSE144240, Mendeley https://doi.org/10.17632/2bh5fchcv6.1, Mendeley https://doi.org/10.17632/svw96g68dv.1, GEO: GSE211895 [http://lifeome.net/supp/livercancer-st/data.htm, https://www.ncbi.nlm.nih.gov/geo/query/acc.cgi?acc=

GSE144240, https://data.mendeley.com/datasets/2bh5fchcv6/1, https://data.mendeley.com/datasets/svw96g68dv/1, https://www.ncbi.nlm.nih.gov/geo/query/acc.cgi?acc=GSE211895, https://www.10xgenomics.com/resources/datasets][59,101–104,110]. The drug response data used in this study are available in the PharmacoDB database [https://pharmacodb.ca/][111]. Source data are provided with this paper.

## Code availability

Software used for analysis is public and described in detail in the Methods section. Raw scripts and code are available at https://github.com/rohitarorayyc/SpatialTranscriptomics/. Citable code for this study is available at the Zenodo https://doi.org/10.5281/zenodo.8079095 (https://zenodo.org/record/8079095)[112].

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

## Acknowledgements

This study was funded by the PRecision Oral Biology (PROBE) internal grant from the Ohlson Research Initiative and Charbonneau Cancer Institute to P.B. We would like to thank the Precision Oncology and Experimental Therapeutics (POET) program for their help with patient consenting and sample acquisition. We thank the Centre for Health Genomics and Informatics (CHGI) for providing the sequencing infrastructure to spatially profile our RNAseq libraries. We would like to thank the University of Calgary High Performance Computing Cluster (ARC) for providing data analysis infrastructure. We thank Danielle Simonot for her help with tissue retrieval from the Alberta Cancer Research Biobank (ACRB). We thank Christina Yang for her help with cryosectioning and slide preparation and H&E staining of profiled samples. We thank Dr. Mayi Arcellana-Panlilio and Dr. Guido van Marle for their extremely kind mentorship and productive feedback. We would like to thank Arzina Jaffer, Leslie Cao,Keerthana Chockalingam for help with bioinformatics consultation. We thank Nicole Rosin for her help with antibody procurement. We are grateful to Dr. Anne Vaahtokari at the Charbonneau Institute Microscopy Facility for assistance with acquiring microscopic images. We would also like to thank the patients and their families for consenting to provide tissue for this study.

## Author contributions

R.A. and C.C. performed data analysis, generated the atlas, prepared figures and contributed to study design. M.K., A.C. and D.S. conducted spatial transcriptomics experiments and collected data. T.W.M., S.C., R.H. and J.C.D. contributed to sample collection and tumor banking. R.M. and R.K.A., contributed to bioinformatics analyses. S.S. performed machine learning analyses, deposited study data, and contributed to atlas generation. J.B. and P.N. oversaw bioinformatics analyses. M.H. performed histopathological annotation of tumor samples and contributed to study design. P.B. conceived the study, and supervised all aspects of research design and bioinformatics analyses. R.A., C.C. and P.B. co-wrote the paper with input from all other authors.

## Competing interests

R.A. was a bioinformatics consultant at Phenomic AI. R.K.A was previously a Venture Fellow at Flagship Pioneering. R.K.A has served as a Technical Consultant for the Bill and Melinda Gates Foundation Strategic Investment Fund and was a former Senior Policy Advisor at Health Canada. R.K.A also holds a minority stake in Alethea Medical. P.B. is the co-founder and Vice President of Management and Planning at Onco-Helix, Inc. All other study authors declare no competing interests.
