## [Peer Review File · Nature Communications]

Spatial transcriptomics reveals distinct and conserved tumor core and edge architectures that predict survival and targeted therapy responseReviewers' Comments:

Reviewer #1:

Remarks to the Author:

In this manuscript, the authors present an analysis of oral squamous cell carcinoma using spatial transcriptomics data, together with integrative analyses with TCGA and other scRNA-Seq datasets. The authors distinguish the TC and LE regions in each tumor, and find that the spots annotated for each region are highly correlated across patients, and have distinct expression profiles. These regions represent different functions in the tumor, as the LE expresses higher EMT signature, which its present correlates with worse overall survival. The manuscript is clearly written and presented. While the analysis of the two distinct regions of the tumor from a spatial and transcriptomic perspective is interesting, it is not a novel observation. Specifically the differences in regional expression may be explained by the presence of spatially unique cancer cell states.

Overall, the authors report on a useful and extensive dataset, however they do not analyze their claims with the appropriate framework of analysis. Specifically, have the following main concerns:

The authors highlight the use of single-cell data integration in the abstract and throughout the manuscript as a central part of the work. However the integration with scRNA seq data is presented only later in the manuscript, and is done by using published HNSCC; i.e. not matched with the ST samples. While this data is appropriately used to deconvolve the ST spots, the way it is presented in the text is misleading.

The authors claim that they find distinct cancer cell states between the LE and TC regions. In fact a central claim of their paper - in the last paragraph of the background - "We find that differences in regional expression may be explained by the presence of spatially unique cancer cell states." However, the evidence for this claim is included only in the extended data (5). Indeed the evidence is not sufficiently comprehensive to qualify as a description of cell states. The authors do have the ability to annotate the 'cancer only' spots and analyze them for their cancer cell states, but currently, this is not really done. The authors do not make any specific conclusions from this analysis.

The differences between the TC and LE regions shown in Figure 1 are impressive because they transcend patients. However, it is revealed in Figure 4 that the differences are largely driven by a combination of different signatures of cancer cells and a different composition of cells. This is a critical point that the authors do not clarify from the outset. Thus, it isn't clear what is the significance of the Figure 1 findings, since they may stem mainly from the presence of CAFs in one of these regions. It should be clear if the differences described in figure 2, for example, are between the regions or between the cancer cells in these regions.

In figure 3E, the LE annotation looks to be highly overlapping with stromal tissue (based on the H&E). This may suggest that the ST data is picking up stromal rich areas across different cancers, but that the cancer core is more tissue specific. If this is the case, this analysis does not add to our current knowledge about tumors and areas within the tumor, as this annotation can be simply done by H&E.

The authors find that higher scores of LE, correlates with worse outcomes. Since this signature seems to be related to the EMT cells, and invasion of the tumor, this is to be expected, as it also correlates with metastasis (Fig. 3j). In other words, if the signature includes EMT related genes, then the result is to be expected given a wealth of previous results. As it stands, a novel finding is not supported regarding the existence of this signature\region.

Based on the ST data, the authors show that it is challenging to score the different tumor types for the TC signature (fig. 3h). It is unclear how did the authors manage to use that score for the TCGA data?

The authors claim that two distinct types of CSC reside in LE and TC, this could be validated in a single cell method (such as immunofluorescence), since the analysis is done on ST slides. Also, it should be tested if these are different clones that contribute to these populations or this is cell plasticity. To

make this point stronger, the authors can distinguish between the stem-like and non stem-like cancer cells in each region, and specify if the differences between the regions are derived only by the differences between the cancer stem cells or also by the rest of the cancer cells in the region.

In figure 5, what does it mean that the trajectories are from TC to LE, while the time scale of RNA velocity is much shorter than the time scale for such a phenotypic change in the tumor?

Minor comments:

In Fig1 a, it seems like there are 10 samples, although the text says 11 patients and 12 samples. Also, these are 12 arrays of ST and not 12 slides.

In figure 2c, the fold change graph, should show all patients for all the genes. It is not clear and might be misleading to show only the patients that follow the general trajectory that the authors want to show for each gene.

Reviewer #2:

Remarks to the Author:

In this study, the investigators performed spatial transcriptomic analysis on oral squamous cell carcinoma (OSCC) to characterize tumor core (TC) and leading edge (LE) transcriptional features. They analyzed transcriptional profiles, cellular compositions, and ligand-receptor interactions comparing TC and LE spots. They showed that spatially-related gene signatures were associated with clinical outcomes and can be seen across different cancer types. They also used in silico modeling to demonstrate that effective drugs and ineffective drugs produced opposite RNA velocity patterns between TC and LE spots. Overall, this is a novel study in terms of technologies and computational analyses. Some of the findings are interesting and the dataset is useful for the research community. However, I have a number of concerns regarding the research design and data analyses and interpretation, as outlined below.

1) The annotation of TC and LE is critically important for the present work, but there are concerns: 1) from the method description, 15 LE genes and 7 TC genes were identified by literature search. However, it is unclear what criteria/cutoffs were applied to select these markers (and even the literature). 2) some of the markers such as FOXP3, CD8A, FN1, COL4A2, were associated with very specific cell types, and they will strongly affect most of the downstream analyses. For example, it is not surprising that they can be separated by the UMAP projection. Also see comments #5. This is a major concern.

2) The authors interpreted that TC and LE differences seemed visually conserved across patients in UMAP projections from Fig.1d-e. It would be much more helpful to plot TC vs LE abundant across each patient, since it is very hard to see at this scale.

3) In the pathway and hallmark analyses (Fig.2b), since many pathways were scanned, p values need to be adjusted for multiple test burden. Also, a complete list of different pathways between TC vs LE will be more informative and will be a good resource for HNSCC community.

4) Related to Fig.2b, are there any pathways/hallmarks higher in TC than LE?

5) Related to #1, the DEG analysis in Fig.2C is actually circular: for example, TNC and FN1 were used as markers to annotate LE spots in the first place, and expectedly it will be identified as top upregulated genes in LE. Reciprocal examples can be seen in TC (SPRR1B, SPRR2D). This circular problem goes beyond just a few genes, since these genes are associated with specific cell types, for

example FN1 is strongly expressed by fibroblasts. This calls into question again regarding the initial annotation, and affects many of the downstream analyses. For example, the observation that LE was generally most enriched for the mesenchymal subtype (Extended Data Fig. 2e) is likely because LE has more fibroblastic/EMT genes when because of the annotation. Relatedly, the mesenchymal subtype is known to have worse survival and thus the survival analyses (Fig.3g) may also be affected.

6) In Fig.3C, in order to show that ML-predicted LE annotations corroborated those annotated by the study authors, LE and TC annotated from the original papers need to be indicated.

7) The dynamo/drug sensitivity analyses are far-stretched and poorly developed. First, there are 220 drugs but only a few examples are shown. Were the results statistically tested? Ie, how confident is conclusion that effective drugs and ineffective drugs produced the opposite RNA velocity patterns? Moreover, what does this mean exactly in terms of cell biology? Why were effective drugs reversing the RNA flow and what is possible mechanistic basis?

8) Some of the most important pieces of data would be much more strengthened by orthogonal validations. For example, the differential cellular compositions shown in Fig.4 can be validated using immunostaining using the same or independent sample cohort.

Reviewer #3:

Remarks to the Author:

In this article, the authors investigate biological mechanisms underlying the prognostic association with the tumor leading edge in oral squamous cell carcinoma (OSCC). Specifically, they seek to leverage spatial transcriptomics and single cell RNA-seq to uncover novel associations with the leading edge and aggressive clinical behaviour. Surprisingly, they find conserved programs across multiple cancer types that dictate leading edge function.

Altogether, this is a fascinating study that adds positively to an emerging and important field. The authors have performed sophisticated analyses with results that largely support their conclusions. The figures are easy to interpret and follow. Strengths include integration between spatial transcriptomic data and existing data from scRNA-seq, TCGA, and other sources. There are exciting findings about the implications of the results on patient prognosis across multiple cancer types as well as future drug discovery efforts.

Weaknesses include the lack of (1) characterization of immune stroma, (2) multi-sample analysis from the same tumor, and (3) validation of drug predictions. While to some extent these analyses could be beyond the scope of this study, the authors are encouraged to consider the specific comments below, which – if addressed – could result in an even stronger manuscript.

MAJOR COMMENTS:

- The literature-curated TC genes include multiple immune genes which would typically be expected to be present in stroma. What is the rationale for including these genes as positive control for TC?

- Figure 1c/e: Are the TC and LE enriched for any of the pathologist-defined segmentations/annotations? Descriptive stats and comparisons would be informative. It would also be informative to compare the pathologist-defined annotations with the scRNAseq cell type analysis presented in Figure 4.

- Figure 2a: A few samples have a LE with similar or even higher correlation to TCs than to other LEs (e.g., S12). Is this related to relative inferred purity of the LE and TC samples for those patients?

- In the TCGA analysis, what is the correlation between TC and LE enrichment scores? Can the authors test whether these are independently prognostic using multivariable Cox models? Or alternatively, are they essentially redundant and anti-correlated features?
- In Figure 4a, can the authors speculate as to why there is not a strong immune stroma component in either the TC or LE (even though there were immune related genes in the literature-curated gene sets)?
- Figure 5 presents potentially exciting results with significant implications for future work. The interpretation is limited by lack of quantification of the vector field effects. The authors are urged to demonstrate quantitatively the magnitude of enrichment for the trajectories depicted in Figure 5g. Without this, it is difficult to know how large and generalizable this effect is among different drugs.
- Are there specific classes of drugs that would be predicted to have the desired effect?
- The in silico drug discovery analysis would be strengthened by further evaluation in other cancer types. Are the drug predictions selective for HPV-negative SCC, or if the authors are correct that the LE programs are shared across multiple cancer types, are there also shared drug predictions for reversing the LE phenotype?
- The in silico drug discovery analysis would be strengthened by validation using preclinical models with drug treatment.

MINOR COMMENTS:

- Figure 1a: The figure indicates that 6/10 patients are M1. Is this accurate? It is not consistent with the overall stage categories of I-IVA. M1 would be stage IVC. Why were so many M1 patients included in this study?
- Extended Data Figure 1: Provide color key for pathologist annotations.
- Figure 4 and Extended Data Figure 5b: Provide key for sample number.

Reviewer #4:

Remarks to the Author:

This manuscript performed spatial single-cell transcriptomics analysis on HPV-negative oral squamous cell (OSCC). Authors for the first time characterized molecular signatures of tumor core (TC) and leading edge (LE) regions and their functional and pathological roles across multiple tumor types. They also predicted their drug response patterns. It is a very interesting and novel work. The manuscript is well-written.

Major issues.

1. More information is needed for the machine learning to predict TC, LE, and other spots. Visualization in Figure 3c-e is helpful but is insufficient to justify the accuracy of prediction. A quantitative measurement such as pointwise uncertainty quantification is expected. Similarly, the confidence of prediction should be estimated for differentiation trajectory predictions.
2. Machine learning models and statistics analyses primarily provide correlations between TC/LE

signatures and prognoses. It will be interesting to know what causal molecular programs are.

Minor issues.

1. It will be helpful to elaborate what therapeutic targets were derived for OSCC in the main text.

Response to Reviewer 1 Comments

Reviewer #1, expertise in scRNAseq/ST and the TME (Remarks to the Author):

In this manuscript, the authors present an analysis of oral squamous cell carcinoma using spatial transcriptomics data, together with integrative analyses with TCGA and other scRNA-Seq datasets. The authors distinguish the TC and LE regions in each tumor, and find that the spots annotated for each region are highly correlated across patients, and have distinct expression profiles. These regions represent different functions in the tumor, as the LE expresses higher EMT signature, which its present correlates with worse overall survival. The manuscript is clearly written and presented. While the analysis of the two distinct regions of the tumor from a spatial and transcriptomic perspective is interesting, it is not a novel observation. Specifically the differences in regional expression may be explained by the presence of spatially unique cancer cell states.

Overall, the authors report on a useful and extensive dataset, however they do analyze their claims with the appropriate framework of analysis. Specifically, have the following main concerns:

1. The authors highlight the use of single-cell data integration in the abstract and throughout the manuscript as a central part of the work. However the integration with scRNA seq data is presented only later in the manuscript, and is done by using published HNSCC; i.e. not matched with the ST samples. While this data is appropriately used to deconvolve the ST spots, the way it is presented in the text is misleading.

We have now restructured our analyses and manuscript to focus on spatially-deconvolved cancer cells within our ST samples by performing integration with published scRNA seq data at the outset of the manuscript. See Figure 1. Furthermore, we have provided additional clarification to the origin of the scRNA-seq dataset used in our analysis.

Page 4: *We next determined the composition of malignant tumor cells and other cellular subpopulations present in the pathologist-annotated squamous cell carcinoma regions by performing integrative analysis of our ST data with a separate, publicly-available HNSCC scRNA-seq dataset¹³*

2. The authors claim that they find distinct cancer cell states between the LE and TC regions. In fact a central claim of their paper - in the last paragraph of the background - "We find that differences in regional expression may be explained by the presence of spatially unique cancer cell states." However, the evidence for this claim is included only in the extended data (5). Indeed the evidence is not sufficiently comprehensive to qualify as a description of cell states. The authors do have the ability to annotate the 'cancer only' spots and analyze them for their cancer cell states, but currently, this is not really done. The authors do not make any specific conclusions from this analysis.

We thank the reviewer for this comment. We have restructured our study to focus on ‘cancer only’ spots based on cellular deconvolution and CNV prediction. Using the combination of these techniques, we are able to identify ‘cancer only’ spots with high confidence.

Page 4: *We next determined the composition of malignant tumor cells and other cellular subpopulations present in the pathologist-annotated squamous cell carcinoma regions by performing integrative analysis of our ST data with a separate, publicly-available HNSCC scRNA-seq dataset.¹³ To identify malignant tumor spots, we stringently characterized malignant cells as having a deconvolution score > 0.99 (Fig. 1c), or CNV probability score >0.99 (Fig. 1d). CNV analysis revealed recurrent deletions in chromosome 3, and amplifications in chromosome 9 (Extended Data Fig. 1n). All 12 samples were identified to have both spatially deconvolved or CNV-inferred cancer cells based on the applied cutoff with high confidence, resulting in 13950 malignant and 10852 nonmalignant spots (Fig. 1e and Extended Data Fig. 1o).*

After annotating for ‘cancer only’ spots, we discovered that transcriptomic differences between the TC and LE could not be explained by differences in HNSCC molecular subtype composition, tumor clonality, and differing cancer stem cell (CSC) proportions. Together with these findings, we propose that the transcriptomic differences are attributed to unique cancer cell states.

Page 7: *Therefore, we believe that the distinct biological profiles of the TC and LE are explained by the presence of unique cancer cell states—conserved gene expression programs that dynamically manifest from specific tumor microenvironment interactions – comprising both CSC and non-stem-like malignant cells.^{53,54} Previous literature exploring dynamic CSC states has proposed the existence of epithelial-like CSCs that inhabit the LE and mesenchymal-like CSCs inhabit the TC (Fig. 3b).⁵⁵ When we integrated gene-sets associated with these distinct CSC states to our ST dataset, we observed higher expression of the mesenchymal-like CSC state in the LE ($p < 0.001$) and epithelial-like CSC state in the TC ($p < 0.001$) (Fig. 3c,d). The localization of these CSC states was further validated through immunofluorescence staining of serial tissue sections, which revealed localization of the CD24 marker at the TC, and the CD44 marker at the LE (Extended Data Fig. 3e). These findings reinforce the plasticity within the TC and LE niches that promote the propagation of transcriptionally unique cancer cell states.*

3. The differences between the TC and LE regions shown in Figure 1 are impressive because they transcend patients. However, it is revealed in Figure 4 that the differences are largely driven by a combination of different signatures of cancer cells and a different composition of cells. This is a critical point that the authors do not clarify from the outset. Thus, it isn't clear what is the significance of the Figure 1 findings, since they may stem mainly from the presence of CAFs in one of these regions. It should be clear if the differences described in figure 2, for example, are between the regions or between the cancer cells in these regions.

As stated earlier, we have restructured our analysis and manuscript to begin by specifically annotating ‘cancer only’ spots. By restricting our analysis to ‘cancer only’ spots, our downstream analyses conclusively determines that the observed differences in the TC and LE are a function of the cancer cells in these regions. Our reanalysis using ‘cancer only’ spots produced similar results to our original work and revealed that the differences between the TC and LE regions transcend patients (Fig. 2e), alleviating prior concerns expressed by the reviewer that the different regional transcriptomic profiles could be attributed to different cellular compositions (e.g., CAFs).

4. In figure 3E, the LE annotation looks to be highly overlapping with stromal tissue (based on the H&E). This may suggest that the ST data is picking up stromal rich areas across different cancers, but that the cancer core is more tissue specific. If this is the case, this analysis does not add to our current knowledge about tumors and areas within the tumor, as this annotation can be simply done by H&E.

As mentioned earlier, we have restructured our analyses and manuscript to begin by specifically annotating ‘cancer only’ spots, therefore ensuring that our downstream annotations do not comprise any stromal or non-malignant components. Most of our original findings have remained unchanged even when analyzing ‘cancer only’ spots, enabling us to expand on the current state of knowledge concerning malignant cell states.

5. The authors find that higher scores of LE, correlates with worse outcomes. Since this signature seems to be related to the EMT cells, and invasion of the tumor, this is to be expected, as it also correlates with metastasis (Fig. 3j). In other words, if the signature includes EMT related genes, then the result is to be expected given a wealth of previous results. As it stands, a novel finding is not supported regarding the existence of this signature\region.

We agree with the reviewer that there was a possibility that our initial analysis containing the LE signature could have been biased from picking up stromal regions that overrepresented EMT-related genes. When generating our “high” LE enrichment score, these genes could incorrectly skew our prognostic findings, and compromise the novelty of our work.

However, our current restructured analysis now rectifies these concerns by ensuring that our LE signature is restricted to malignant spots. Our findings now confirm that the LE is a unique tumoral niche that demonstrates several invasion-related properties, which have not been previously described. We provide further evidence that our signature is not solely biased by EMT related programs through our correlation analysis with a previously identified CAF signature. CAFs have historically been recognized as prognostic due to their involvement in ECM-remodeling and expression of EMT-related programs.¹ Our findings ultimately show that other mechanisms beyond EMT may be at play in guiding the prognostic features associated with the LE.

Page 10: *A weak negative correlation was also observed between EPIC CAF and our LE signature enrichment scores ($r=-0.23$, $p<0.05$) (Extended Data Fig. 5f), which may suggest that there are other unexplored mechanisms beyond CAF activity driving survival outcomes associated with the LE.*

6. Based on the ST data, the authors show that it is challenging to score the different tumor types for the TC signature (fig. 3h). It is unclear how did the authors manage to use that score for the TCGA data?

We acknowledge that by applying the TC signature to other cancers, we conflict with the findings from our ML model. We have now corrected our pan-cancer prognostic analysis to only examine the prognostic ability of the LE signature (Fig. 5c, d), and exclude the TC signature due to its tissue-specific nature.

We still show that the TC signature is associated with patient outcomes in oral cancers in the TCGA data (Fig. 5b) and an independent validation cohort (Extended Data Fig. 5c). We can score the TC signature in these datasets as they are the same cancer type (oral cancer) as the data used in our ST analysis.

7. The authors claim that two distinct types of CSC reside in LE and TC, this could be validated in a single cell method (such as immunofluorescence), since the analysis is done on ST slides. Also, it should be tested if these are different clones that contribute to these populations or this is cell plasticity. To make this point stronger, the authors can distinguish between the stem-like and non-stem-like cancer cells in each region, and specify if the differences between the regions are derived only by the differences between the cancer stem cells or also by the rest of the cancer cells in the region.

We have now performed immunofluorescence staining of epithelial-like and mesenchymal-like CSC state markers (CD24 and CD44, respectively) in serial sections to validate their expression in OSCC tissue (Extended Data Fig. 3e).

We have also performed an additional clonal analysis using the Numbat package in our investigation of TC and LE cancer cell states and find that proportions of subclones do not differ across different tumoral spatial regions (Extended Data Fig. 3d), reinforcing that TC and LE differences are likely attributed to cellular plasticity.

Page 6: *Next, we considered the association between tumor subclonal architectures and the OSCC TC and LE by inferring clonal lineages and evolutionary history through CNV events with the Numbat package. We found that multiple subclonal lineages were present throughout the OSCC tumor, with similar proportions of subclonal populations across TC and LE regions (Extended Data Fig. 3d).*

Further, we show that CSCs are found throughout the tumor architecture along with non-stem-like malignant cells, irrespective of spatial regions (Fig. 3a). Therefore, we believe that

differences in the LE and TC are attributed to unique cancer cell states that encompass both CSC and non-stem-like malignant cells.

Page 7: *We then asked if the contribution of cancer stem cells (CSCs) could help explain the differences in TC and LE expression profiles. CSCs are cancer cell populations that possess stem-cell like progenitor and malignant properties.⁵¹ Given the abundance of EMT-related, metastatic, and invasive expression programs at the LE, we hypothesized that CSCs may be preferentially localized in the LE. However, we found no significant differences in the expression of canonical OSCC CSC markers⁵² between the LE and TC ($p>0.05$) (Fig. 3a). Furthermore, expression of CSC markers was seen evenly throughout UMAP projections, indicating that CSC populations are found throughout the OSCC tumor (Fig. 3a; density plot) along with non-stem-like malignant cells.*

Therefore, we believe that the distinct biological profiles of the TC and LE are explained by the presence of unique cancer cell states—conserved gene expression programs that dynamically manifest from specific tumor microenvironment interactions – comprising both CSC and non-stem-like malignant cells.^{53,54}

We also provide additional clarification that there is an enrichment for two distinct types of CSC states in the LE and TC, rather than distinct CSC cell populations. The CSC states and associated markers have been previously identified as plastic, and change in response to the tumor microenvironment.² These findings help provide further evidence that the observed differences between the TC and LE regions can be explained by the existence of unique cancer cell states.

Page 7: *Previous literature exploring dynamic CSC states has proposed the existence of epithelial-like CSCs that inhabit the LE and mesenchymal-like CSCs inhabit the TC (Fig. 3b).⁵⁵ When we integrated gene-sets associated with these distinct CSC states to our ST dataset, we observed higher expression of the mesenchymal-like CSC state in the LE ($p<0.001$) and epithelial-like CSC state in the TC ($p<0.001$) (Fig. 3c,d). The localization of these CSC states was further validated through immunofluorescence staining of serial tissue sections, which revealed localization of the CD24 marker at the TC, and the CD44 marker at the LE (Extended Data Fig. 3e). These findings reinforce the plasticity within the TC and LE niches that promote the propagation of transcriptionally unique cancer cell states.*

8. In figure 5, what does it mean that the trajectories are from TC to LE, while the time scale of RNA velocity is much shorter than the time scale for such a phenotypic change in the tumor?

We thank the reviewer for this important question. In our study, we wish to clarify that instead of using the original approach to deriving RNA velocity by La Manno et al.³, we utilized scVelo.⁴ scVelo, a dynamical modeling-based iteration of RNA Velocity, extends the applicability of RNA velocity estimation to transient systems and systems with heterogeneous subpopulation kinetics by incorporating the dynamics of gene expression changes over time. Unlike traditional RNA

velocity estimation methods, which rely on steady-state assumptions and assume linear and constant gene expression changes over time, scVelo utilizes ordinary differential equations (ODEs) to capture the time-dependent dynamics of gene expression changes. This allows scVelo to model changes in gene expression over processes such as cellular differentiation or response to external stimuli, as well as account for heterogeneous subpopulation kinetics where distinct cells may exhibit different gene expression dynamics. In their original paper, the authors utilized scVelo to disentangle subpopulation kinetics in neurogenesis and pancreatic endocrinogenesis, both of which involve observed phenotypic changes over extended time frames. Building on this success, we and others have effectively employed scVelo to investigate diverse biological questions. For example, we previously used scVelo to interrogate wound-responsive fibroblast trajectories spanning multiple biological days (Sinha, Sparks et al., Cell 2022), while Li et al. employed scVelo to derive "normal to tumor-associated macrophage trajectories", similar to the tumor core to tumor edge transitions outlined in our manuscript.

With regards to the specific query about Figure 5, the observed trajectories depict the inferred paths of cells within the tumor, from their initial state to their final state, as they transition from the tumor core state towards the leading edge state but does not refer to the time scale for tumor phenotypic change. Rather, the time scale referenced in Figure 5 refers to the time scale relevant for gene specific differences in RNA velocity. We regret any confusion caused using the term "RNA velocity" in the figure. We provide additional clarification to the rationale behind the time scale vector in our new Figure 6.

Minor comments:

1. In Fig1 a, it seems like there are 10 samples, although the text says 11 patients and 12 samples. Also, these are 12 arrays of ST and not 12 slides.

Thank you for highlighting this. Our analysis had 12 samples, and the 11 patients were a typo on our end. We have corrected Fig. 1a to indicate 10 patients now correctly. We chose to continue using the term "slide" in place of arrays, as the 10X genomics technology utilizes spotted arrays found on the surface of glass slides.

2. In figure 2c, the fold change graph, should show all patients for all the genes. It is not clear and might be misleading to show only the patients that follow the general trajectory that the authors want to show for each gene.

Our fold change graphs display genes differentially expressed in at least 9 out of 12 patients, but all patients are faithfully shown on the graph. Interestingly, there are no patients which do not follow the general trajectory and we are not omitting any patients from the fold change graphs. For more information, please refer to Supplementary Table 5 which reports DEG Log2FC values, stratified by patient.

Response to Reviewer 2 Comments

Reviewer #2, expertise in OSCC genomics (Remarks to the Author):

1. In this study, the investigators performed spatial transcriptomic analysis on oral squamous cell carcinoma (OSCC) to characterize tumor core (TC) and leading edge (LE) transcriptional features. They analyzed transcriptional profiles, cellular compositions, and ligand-receptor interactions comparing TC and LE spots. They showed that spatially-related gene signatures were associated with clinical outcomes and can be seen across different cancer types. They also used in silico modeling to demonstrate that effective drugs and ineffective drugs produced opposite RNA velocity patterns between TC and LE spots. Overall, this is a novel study in terms of technologies and computational analyses. Some of the findings are interesting and the dataset is useful for the research community. However, I have a number of concerns regarding the research design and data analyses and interpretation, as outlined below.

1) The annotation of TC and LE is critically important for the present work, but there are concerns: 1) from the method description, 15 LE genes and 7 TC genes were identified by literature search. However, it is unclear what criteria/cutoffs were applied to select these markers (and even the literature). 2) some of the markers such as FOXP3, CD8A, FN1, COL4A2, were associated with very specific cell types, and they will strongly affect most of the downstream analyses. For example, it is not surprising that they can be separated by the UMAP projection. Also see comments #5. This is a major concern.

Thank you for highlighting this. We have now addressed these concerns by restructuring our analysis, please refer to Reviewer 2, Major Comment #5 for more details.

2) The authors interpreted that TC and LE differences seemed visually conserved across patients in UMAP projections from Fig.1d-e. It would be much more helpful to plot TC vs LE abundant across each patient, since it is very hard to see at this scale.

Thank you for highlighting this. We have now added this abundance plot in Extended Data Fig. 2m.

3) In the pathway and hallmark analyses (Fig.2b), since many pathways were scanned, p values need to be adjusted for multiple test burden. Also, a complete list of different pathways between TC vs LE will be more informative and will be a good resource for HNSCC community.

We had performed a Benjamini-Hochberg adjustment for the multiple tests performed in our hallmark pathway analyses. We have now updated our manuscript text and methods to reflect this additional analysis.

Page 5: *We found that LE spots displayed higher expression of genes associated with cell cycle ($p\text{-adj}<0.001$), epithelial-mesenchymal transition (EMT) ($p\text{-adj}<0.001$), and angiogenesis ($p\text{-adj}<0.001$) (Extended Data Fig. 2n).*

Page 16: *Select hallmark cancer gene signatures were scored in each spot using the Seurat (Version 4.3.0) function 'addmoduleScore'. To test for differences in the module scores calculated between the TC and LE, a two-sided paired Wilcoxon Rank sum test was conducted using the 'ggpubr' (Version 0.5.0) function 'stat_compare_means' and corrected using a Bonferroni correction.*

We have also generated a comprehensive list of different pathways that were enriched in the TC and LE. We have included these findings in Extended Data Fig. 2o, and supplementary table 3

Page 15-16: *Differentially regulated hallmark pathways between core and edge cancer cell states were identified by modifying code from the SCPA R package (Version 1.2.0). The 'compare_seurat' function queried hallmark genesets in Seurat (Version 4.3) R objects housing integrated data and tested for differential pathway activity using multivariate distribution testing with a Bonferroni correction applied. Plots comparing hallmark genesets were created using the 'ggplot2' (Version 3.4.0) and 'ggrepel' R packages (Version 0.9.2).*

4) Related to Fig.2b, are there any pathways/hallmarks higher in TC than LE?

We have now outlined additional pathways that are enriched in the TC, relative to the LE within our manuscript text.

Page 5: *Cellular function hallmarks that were upregulated in the TC included keratinization, cell differentiation, as well as antimicrobial and immune-related pathways...*

Furthermore, as stated earlier, we have included a comprehensive list of different pathways that were enriched in the TC and LE (Extended Data Fig. 2o, and supplementary table 3)

5) Related to #1, the DEG analysis in Fig.2C is actually circular: for example, TNC and FN1 were used as markers to annotate LE spots in the first place, and expectedly it will be identified as top upregulated genes in LE. Reciprocal examples can be seen in TC (SPRR1B, SPRR2D). This circular problem goes beyond just a few genes, since these genes are associated with specific cell types, for example FN1 is strongly expressed by fibroblasts. This calls into question again regarding the initial annotation, and affects many of the downstream analyses. For example, the observation that LE was generally most enriched for the mesenchymal subtype (Extended Data Fig. 2e) is likely because LE has more fibroblastic/EMT genes when because of the annotation. Relatedly, the mesenchymal subtype is known to have worse survival and thus the survival analyses (Fig.3g) may also be affected.

We appreciate how the reviewer might think that our analysis is circular and that some of the marker genes used for the TC and LE annotations were included in the DEGs. We have now performed a complete reanalysis of our data and have restructured our analysis to begin by annotating 'cancer only' spots, followed by an unsupervised analysis to identify 3 major clusters, which were later labeled as TC, transitory, and LE regions for further analysis. By annotating our

tumoral regions with an unsupervised analysis, we ensure that our downstream analyses are no longer biased by our initial annotations.

Page 5: *After identifying and annotating malignant tumor spots that were primarily composed of cancer cells, we performed unsupervised louvain-clustering to unravel the spatial heterogeneity in cancer cell expression profiles. We generated 14 louvain clusters among aggregated malignant spots that could be partitioned into 3 major clusters (Fig. 2a). We then characterized the major clusters through differential gene expression analysis (DGEA) (Fig. 2b). Top DEGs enriched in cluster 1 included genes involved in keratinization SPRR2D, SPRR2E, SPRR2A, and inhibition of EMT DEFB4A and LCN2 (ref.^{18,19}), while DEGs in cluster 3 included genes involved in the ECM matrix COL1A1, FN1, COL1A2, TIMP1, COL6A2 (Fig. 2b and Supplementary Table 2). DEGs enriched in cluster 2 shared attributes of both cluster 1 and 3, with genes involved in keratinization KRT6C, KRTDAP, KRT6B (ref.²⁰), and ECM remodeling LYPD3, SLPI (ref.^{21,22}) (Fig. 2b and Supplementary Table 2). Interestingly, the expression of CLDN4 and SPRR1B HNSCC TC markers,¹³ and LAMC2 and ITGA5 HNSCC LE markers¹³ corresponded to clusters 1 and 3, respectively (Fig. 2b,c and Supplementary Table 2). These findings prompted us to annotate cluster 1 as “tumor core” (TC) and cluster 3 as “leading edge” (LE). Cluster 2 was annotated as “transitory” due its composition of TC and LE DEG programs (Fig. 2d and Extended Data Fig. 2a-m).*

6) In Fig.3C, in order to show that ML-predicted LE annotations corroborated those annotated by the study authors, LE and TC annotated from the original papers need to be indicated.

Thank you for this point. As the exact annotations of the leading edge annotations were not made publicly accessible by the study authors for quantitative evaluation, we opted to remove this claim from our manuscript. In turn, we solidified the power of our ML-prediction algorithm through 10-fold cross validation on our own data, highlighting a high predictive capability (Figure 4c and Supplementary table 7).

7) The dynamo/drug sensitivity analyses are far-stretched and poorly developed. First, there are 220 drugs but only a few examples are shown. Were the results statistically tested? Ie, how confident is conclusion that effective drugs and ineffective drugs produced the opposite RNA velocity patterns? Moreover, what does this mean exactly in terms of cell biology? Why were effective drugs reversing the RNA flow and what is possible mechanistic basis?

Thank you for your comment and we understand that our previous analysis could be strengthened. To analyze more drugs than the few examples shown, we have now integrated our cell line efficacy data with a drug-gene interaction database (DGldb), to enable systematic querying of drugs. Through quantifying our *in silico*-based approach using a vector field integration approach, we are now able to perform statistical tests to compare vector field fate probabilities between TC, transitory, and LE cancer cells. We also provide a likely explanation as to why effective drugs can better reverse RNA flow.

Page 11: *Dynamo is an in-silico technique that is capable of accurately predicting cell-fate transition following genetic perturbation based on learned splicing vector fields.⁶⁸ We analyzed PharmacoDB drug-response data⁶⁹ for 417 drugs across at least 25 HPV negative HNSCC cell lines and found 140 drugs with sufficient drug-gene interactions identified as being upregulated or downregulated from the DGIdb⁷⁰ (Fig. 6f and Supplementary Table 9). We then restricted our analysis to 70 drugs that demonstrated significant perturbations, and stratified drugs as ‘high AAC’ and ‘low AAC’ by the median (0.164; 0.026-0.560 [total range]) (Supplementary Table 9). We derived quantitative inferences for net measures of ‘incoming’ and ‘outgoing’ transition probabilities among tumor core and leading edge cells (Fig. 6e). Among effective drugs (high AAC), dynamo based in-silico perturbations in high AAC drugs generally displayed transition hierarchies with a reversal of the baseline state, represented by an increase in outgoing transition probabilities from the LE (Fig. 6g). This was reflected through a significant increase in the quantitative measure of net outgoing LE transition probabilities in high AAC drugs relative to low AAC drugs ($p < 0.05$) (Fig. 6h).*

Page 13: *We hypothesize that effective anticancer drugs direct the transition from a LE-like cancer cell state to a TC-like cancer cell state, therefore making these cells less phenotypically less aggressive by suppressing invasive/metastatic signaling.*

8) Some of the most important pieces of data would be much more strengthened by orthogonal validations. For example, the differential cellular compositions shown in Fig.4 can be validated using immunostaining using the same or independent sample cohort.

We have now included additional orthogonal validations, with immunofluorescence staining of serial tissue sections with CD24 and CD44 markers to confirm the presence of dynamic CSC states (Extended Data Fig. 3e). We have now restructured our manuscript to focus on ‘cancer only’ spots, therefore making our previous differential cellular compositions analysis less relevant to the major findings of our work.

Response to Reviewer 3 Comments

Reviewer #3, expertise in OSCC genomics (Remarks to the Author):

In this article, the authors investigate biological mechanisms underlying the prognostic association with the tumor leading edge in oral squamous cell carcinoma (OSCC). Specifically, they seek to leverage spatial transcriptomics and single cell RNA-seq to uncover novel associations with the leading edge and aggressive clinical behaviour. Surprisingly, they find conserved programs across multiple cancer types that dictate leading edge function.

Altogether, this is a fascinating study that adds positively to an emerging and important field. The authors have performed sophisticated analyses with results that largely support their conclusions. The figures are easy to interpret and follow. Strengths include integration between spatial transcriptomic data and existing data from scRNA-seq, TCGA, and other sources. There

are exciting findings about the implications of the results on patient prognosis across multiple cancer types as well as future drug discovery efforts.

Weaknesses include the lack of (1) characterization of immune stroma, (2) multi-sample analysis from the same tumor, and (3) validation of drug predictions. While to some extent these analyses could be beyond the scope of this study, the authors are encouraged to consider the specific comments below, which – if addressed – could result in an even stronger manuscript.

MAJOR COMMENTS:

1. The literature-curated TC genes include multiple immune genes which would typically be expected to be present in strom. What is the rationale for including these genes as positive control for TC?

Thank you for highlighting this. We have now performed a complete reanalysis of our data and have restructured our analysis to begin by annotating ‘cancer only’ spots, followed by an unsupervised analysis to identify 3 major clusters, which were later labeled as tumor core, transitory, and leading edge regions. By performing an unsupervised analysis, we no longer run into issues with our choice of marker genes for the TC and LE.

2. Figure 1c/e: Are the TC and LE enriched for any of the pathologist-defined segmentations/annotations? Descriptive stats and comparisons would be informative. It would also be informative to compare the pathologist-defined annotations with the scRNAseq cell type analysis presented in Figure 4.

Our TC and LE regions are limited to the pathologist-defined OSCC tumor regions, labeled SCC (Extended Data Fig. 1a-m). We have provided additional clarification in our manuscript to reflect these steps in our analysis.

Page 4: *We next determined the composition of malignant tumor cells and other cellular subpopulations present in the pathologist-annotated squamous cell carcinoma regions by performing integrative analysis of our ST data with a separate, publicly-available HNSCC scRNA-seq dataset.¹³ To identify malignant tumor spots, we stringently characterized malignant cells as having a deconvolution score > 0.99 (Fig. 1c), or CNV probability score > 0.99 (Fig. 1d). All 12 samples were identified to have both spatially deconvolved or CNV-inferred cancer cells based on the applied cutoff with high confidence, resulting in 13950 malignant and 10852 nonmalignant spots (Fig. 1e and Extended Data Fig. 1n).*

Page 5: *After identifying and annotating malignant tumor spots that were primarily composed of cancer cells, we performed unsupervised louvain-clustering to unravel cancer cell expression heterogeneity. We generated 14 louvain clusters among aggregated malignant spots that could be partitioned into 3 major clusters (Fig. 2a).*

Page 5: *These findings prompted us to annotate cluster 1 as “tumor core” (TC) and cluster 3 as “leading edge” (LE). Cluster 2 was annotated as “transitory” due its composition of TC and LE DEG programs (Fig. 2d and Extended Data Fig. 2a-m).*

3. Figure 2a: A few samples have a LE with similar or even higher correlation to TCs than to other LEs (e.g., S12). Is this related to relative inferred purity of the LE and TC samples for those patients?

Thank you for your comment. We did not intend to use the correlation matrix as a marker of LE/TC purity, but to rather explore if our annotated TC and LE regions were conserved across patients based on transcriptomic similarity. We recognize that through our original supervised annotation approach, we may have unintentionally captured differing degrees of TC/LE sample purity. Therefore, we have reframed our analysis to perform an unsupervised annotation approach. We find that our general trends continue to indicate that the TC and LE are conserved across patients.

Page 5: *We then sought to determine whether the patterns of gene expression in the LE and TC were conserved across different patients. To do this, a correlation matrix was generated from the whole transcriptome gene expression profiles within the two spatial regions (Fig. 2ea). A high degree of correlation was generally observed within the TC, and within the LE, across different patients. Interestingly, the correlation between the TC and LE expression programs within each patient was relatively low, highlighting the distinct nature of these compartments in the tumor microenvironment. Therefore, our TC and LE annotations represent regions with unique transcriptomic profiles that are conserved across patients.*

We believe that few samples may show similar or even higher correlation to TCs than to other LEs, and vice versa, due to intratumoral heterogeneity. However, across almost all samples, the correlation between TC-TC and LE-LE exceeds the correlation between TC-LE.

4. In the TCGA analysis, what is the correlation between TC and LE enrichment scores? Can the authors test whether these are independently prognostic using multivariable Cox models? Or alternatively, are they essentially redundant and anti-correlated features?

Thank you for your comment. We have now included additional analysis to directly measure the correlation between TC and LE enrichment scores. We find that these two signatures are very weakly correlated to one another, and believe these findings reinforce that the TC and LE signatures are two distinct programs with important implications on OSCC and cancer prognosis.

Page 9: *TC and LE signatures were also very weakly negatively correlated to one another ($r=-0.17$, $p<0.05$) (Extended Data Fig. 5d).*

5. In Figure 4a, can the authors speculate as to why there is not a strong immune stroma component in either the TC or LE (even though there were immune related genes in the literature-curated gene sets)?

Thank you for your comment. We have now reframed our analysis to focus solely on 'cancer only' spots, which removes concerns of the lack of immune stromal signature in the annotated TC and LE regions. However, based on your comment, we now quantify the number of neighboring immune cells and interactions that the TC and LE regions have.

Page 8: *To further characterize the tumor microenvironment, we identified and quantified the number of adjacent nonmalignant spots neighboring our malignant TC and LE spots. Nonmalignant neighboring spots were approximated as a specific cell type based on the most enriched non-cancer cell after deconvolution. Our analysis found significantly higher numbers of neighboring spots enriched for cytotoxic CD8(+) T cell ($p\text{-adj}<0.01$), *ecm.myCAF* ($p\text{-adj}<0.001$), intermediate fibroblast ($p\text{-adj}<0.01$), and macrophage cells ($p\text{-adj}<0.01$), neighboring LE spots, relative to TC spots (Fig. 3i).*

6. Figure 5 presents potentially exciting results with significant implications for future work. The interpretation is limited by lack of quantification of the vector field effects. The authors are urged to demonstrate quantitatively the magnitude of enrichment for the trajectories depicted in Figure 5g. Without this, it is difficult to know how large and generalizable this effect is among different drugs.

Thank you for your comment and we understand that our previous analysis could be strengthened. To analyze more drugs than the few examples shown, we have now integrated our cell line efficacy data with a drug-gene interaction database (DGldb), to enable systematic querying of drugs. Through quantifying our *in silico* based approach using a vector field integration approach, we are now able to perform statistical tests to compare vector field fate probabilities between TC, transitory, and LE cancer cells.

Page 11: *Dynamo is an in-silico technique that is capable of accurately predicting cell-fate transition following genetic perturbation based on learned splicing vector fields.⁶⁸ We analyzed PharmacoDB drug-response data⁶⁹ for 417 drugs across at least 25 HPV negative HNSCC cell lines and found 140 drugs with sufficient drug-gene interactions identified as being upregulated or downregulated from the DGldb⁷⁰ (Fig. 6f and Supplementary Table 9). We then restricted our analysis to 70 drugs that demonstrated significant perturbations, and stratified drugs as 'high AAC' and 'low AAC' by the median (0.164; 0.026-0.560 [total range]) (Supplementary Table 9). We derived quantitative inferences for net measures of 'incoming' and 'outgoing' transition probabilities among tumor core and leading edge cells (Fig. 6e). Among effective drugs (high AAC), dynamo based in-silico perturbations in high AAC drugs generally displayed transition hierarchies with a reversal of the baseline state, represented by an increase in outgoing transition probabilities from the LE (Fig. 6g). This was reflected through a significant increase in the quantitative measure of net outgoing LE transition probabilities in high AAC drugs relative to low AAC drugs ($p<0.05$) (Fig. 6h).*

7. Are there specific classes of drugs that would be predicted to have the desired effect?

We now stratify our *in silico* based analysis on drug classes categorized based on their mechanism of action. Although we do find significance in quantitative differences of these vector field perturbations, many of our drug classes are composed of only two or three therapeutics, limiting the conclusions we can draw from this analysis (Extended Data Fig. 6e,f).

Page 11: *These findings were further confirmed following drug class stratification, which found significant differences in LE outgoing and TC incoming transition probabilities (Extended Data Fig. 6e,f). However, several drug classes were underpowered to derive conclusive biological inferences (Extended Data Fig. 6e,f and Supplementary Table 9).*

8. The *in silico* drug discovery analysis would be strengthened by further evaluation in other cancer types. Are the drug predictions selective for HPV-negative SCC, or if the authors are correct that the LE programs are shared across multiple cancer types, are there also shared drug predictions for reversing the LE phenotype?

As our ML-classifier model found that the tumor core was only observed in some cancers, a region that is critical in our RNA velocity differentiation hierarchy (and subsequent dynamo gene perturbation drug response), we opted to limit our *in silico* drug discovery analysis to only HPV-negative oral cancer cells. Successfully testing the *in silico* drug discovery analysis would require a complete re-analysis in other cancer contexts, and would ultimately fall beyond the scope of our analysis.

9. The *in silico* drug discovery analysis would be strengthened by validation using preclinical models with drug treatment.

We have opted to refrain from performing additional experimentation involving drug treatment on preclinical models. This is because we felt that this would fall beyond the scope of our exploratory work. Furthermore, the drugs identified from the PharmacDB 2.0 database have been previously tested in OSCC models, and have pre-existing data indicating drug success or failure. Our work primarily sought to identify underlying mechanisms for drug prediction.

MINOR COMMENTS:

1. Figure 1a: The figure indicates that 6/10 patients are M1. Is this accurate? It is not consistent with the overall stage categories of I-IVA. M1 would be stage IVC. Why were so many M1 patients included in this study?

Thank you for highlighting this. This was a mistake in our manuscript. We have now correctly indicated that there are 10 patients that all had M0 staging.

2. Extended Data Figure 1: Provide color key for pathologist annotations.

We have now included a color key for pathologist annotations (Fig. 1b and Extended Data Fig. 1a-m).

3. Figure 4 and Extended Data Figure 5b: Provide key for sample number.

We have now ensured that a key is provided for the sample number in all relevant figures.

Response to Reviewer 4 Comments

Reviewer #4, expertise in in silico modelling, drug response and machine learning (Remarks to the Author):

This manuscript performed spatial single-cell transcriptomics analysis on HPV-negative oral squamous cell (OSCC). Authors for the first time characterized molecular signatures of tumor core (TC) and leading edge (LE) regions and their functional and pathological roles across multiple tumor types. They also predicted their drug response patterns. It is a very interesting and novel work. The manuscript is well-written.

Major issues.

1. More information is needed for machine learning to predict TC, LE, and other spots. Visualization in Figure 3c-e is helpful but is insufficient to justify the accuracy of prediction. A quantitative measurement such as pointwise uncertainty quantification is expected. Similarly, the confidence of prediction should be estimated for differentiation trajectory predictions.

Thank you for your comment. To support the accuracy of our classifier model, we now perform 10-fold cross validation of the classifier within our dataset. Similar to a previous published article also using scPred,⁵ we now report the ROC, sensitivity, and specificity of each model used for each tumoral region (Fig. 4c and Supplementary table 7).

We now include confidence estimates for our initial velocity differentiation trajectories overlaid on a UMAP diagram and find that each spot exhibits a high level of confidence (Fig. 6a).

Page 10: *Among spatially deconvolved cancer cells aggregated across all samples, we observed a differentiation hierarchy originating from TC extending towards LE (Fig. 6a). This hierarchy was highly reproducible and displayed high levels of agreement across spots, reflected by high spot velocity vector field confidence of greater than 0.85 in all spots (Fig. 6a).*

2. Machine learning models and statistics analyses primarily provide correlations between TC/LE signatures and prognoses. It will be interesting to know what causal molecular programs are.

We have now explored several causal molecular programs by analyzing transcription factors and molecular upstream regulators that have predicted activation in the TC and LE from SCENIC and Ingenuity Pathway Analysis, respectively (Extended Data Fig. 2p,q).

Page 6: *We next explored regulatory differences between the TC and LE using single-cell regulatory network inference and clustering (SCENIC) to infer transcription factor (TF) activity. SCENIC analysis identified the upregulation of several proto-oncogenic TFs EGR3 and DLX5 (ref.^{31,32}), and tumor suppressor TFs MXI1, GRHL3, and PITX1 (ref.³³⁻³⁵) in the TC (Extended Data Fig. 2pb and Supplementary Table 3). Conversely, the upregulation of several TFs including cellular development and differentiation regulatory genes TP63 and HOXB2 (ref.^{36,37}), and EMT regulatory genes CREB3L1, TCF4, and NFATC4 (ref.³⁸⁻⁴⁰) were observed in the LE (Extended data Fig. 2pb and Supplementary Table 3). IPA upstream regulatory analysis similarly predicted activation of several proto-oncogenic TFs EHF and BCL3 in the TC (ref.^{41,42}), and EMT regulatory genes SORL1 and EGFR in the LE (Extended data Fig. 2q) (ref.^{43,44}).*

A comprehensive list of SCENIC transcription factors can be accessed via Supplementary Table 4.

Minor issues.

1. It will be helpful to elaborate what therapeutic targets were derived for OSCC in the main text.

We now explore some of the therapeutic targets within the main text, and additionally provide a supplementary table with all information regarding explored therapeutic targets (Supplementary table 9).

Page 13: *Alvocidib is a CDK inhibitor currently under investigation for its use in acute myeloid lymphoma (AML),⁷⁵ which demonstrated above average outgoing LE and incoming TC transitory signals in our analysis. As CDK inhibitors have been previously associated with promising results in OSCC cell lines,^{76,77} Alvocidib may be a promising candidate for further research.*

References

1. Lu S, Hua J, Xu J, Wei M, Liang C, Meng Q, et al. Turning towards nonimmunoreactive tumors: Evaluation of cancer-associated fibroblasts enables prediction of the immune microenvironment and treatment sensitivity in pancreatic cancer. *Comput Struct Biotechnol J*. 2022;20:3911–23.
2. Liu S, Cong Y, Wang D, Sun Y, Deng L, Liu Y, et al. Breast Cancer Stem Cells Transition between Epithelial and Mesenchymal States Reflective of their Normal Counterparts. *Stem Cell Rep*. 2014 Jan;2(1):78–91.
3. La Manno G, Soldatov R, Zeisel A, Braun E, Hochgerner H, Petukhov V, et al. RNA velocity of single cells. *Nature*. 2018 Aug;560(7719):494–8.
4. Bergen V, Lange M, Peidli S, Wolf FA, Theis FJ. Generalizing RNA velocity to transient cell states through dynamical modeling. *Nat Biotechnol*. 2020 Dec;38(12):1408–14.
5. Sinha S, Sparks HD, Labit E, Robbins HN, Gowing K, Jaffer A, et al. Fibroblast inflammatory priming determines regenerative versus fibrotic skin repair in reindeer. *Cell*. 2022 Dec;185(25):4717-4736.e25.

Reviewers' Comments:

Reviewer #2:

Remarks to the Author:

In the revised manuscript, the authors have adequately addressed all my prior concerns. I particularly appreciate that the authors have completed re-analyzed the cancer cell populations via unsupervised clustering followed by differential gene expression. This has effectively removed all of the questions due to pre-defined marker gene-based annotations (e.g., potential bias, circular problem).

In fact, I am very impressed by the fact that this unsupervised clustering approach largely validated the initial marker gene-based approach, strongly suggesting the robustness of the results and its high reproducibility!

I further applaud the authors for providing additional orthogonal validations, with immunofluorescence staining of serial tissue sections with CD24 and CD44 markers to confirm the presence of dynamic CSC states. In addition, their work on drug discovery and integration of the RNA flow of cancer cell state has also been strengthened. I would note that this drug discovery part is still exploratory in nature, however, the analytical design and data mining is sound and, it will stimulate the field to contemplate these data and indications.

Overall, this is an exciting and novel work, with significant implications in the genomics of HNSCC.

Comments on behalf of Reviewer #3

Major Comment

- 1) The Major Comment-1 from Reviewer-3 shares the similar concern with the Comment-5 from Reviewer-2, questioning the use of literature-curated genes in the annotation of tumor spots. During the revision, the authors have performed a completely new "de novo" data analysis which has addressed this question with satisfaction, validating also their initial observations and conclusions.
- 2) The Major Comment-2 has been addressed by the authors.
- 3) For the Major Comment-3, the authors have now performed unsupervised clustering, which confirms their original supervised data. They have also provided explanations regarding a few samples showing the opposite correlation trends, which I agree.
- 4) The Major Comment-4 has been addressed by the authors.
- 5) Regarding the Major Comment-5, the authors have addressed this by identifying and quantifying the number of adjacent nonmalignant spots neighboring either malignant TC or LE spots.
- 6) Addressing the Major Comment-5, the authors have now performed statistical quantification of the vector field effects.
- 7) The authors have provided reasonable exploration and discussion to answer the Major Comment-7.
- 8) Regarding the Major Comment-8, it is appropriate that the scope of such in silico drug discovery is restricted to HPV- OSCC for this current study.
- 9) For Major Comment-9, the authors response is acceptable: some of the drugs identified have been previously evaluated in OSCC models using orthogonal approaches, and it is not imperative for the current study.

Minor Comments

The authors have addressed all three of the Minor Comments.

Reviewer #4:

Remarks to the Author:

All major issues raised have been addressed.

Reviewer #5:

Remarks to the Author:

I have reviewed the revision submitted by the authors. In response to the first round of reviews the authors focussed their spatial transcriptomics analysis on 'cancer only' spots and I believe that this solves many of the problems from the original version. The authors have also added several important validation experiments that bolster the confidence in the results. The work has many interesting analyses and datasets that will be important for the community.

REVIEWERS' COMMENTS

Reviewer #2 (Remarks to the Author):

In the revised manuscript, the authors have adequately addressed all my prior concerns. I particularly appreciate that the authors have completed re-analyzed the cancer cell populations via unsupervised clustering followed by differential gene expression. This has effectively removed all of the questions due to pre-defined marker gene-based annotations (e.g., potential bias, circular problem).

In fact, I am very impressed by the fact that this unsupervised clustering approach largely validated the initial marker gene-based approach, strongly suggesting the robustness of the results and its high reproducibility!

I further applaud the authors for providing additional orthogonal validations, with immunofluorescence staining of serial tissue sections with CD24 and CD44 markers to confirm the presence of dynamic CSC states. In addition, their work on drug discovery and integration of the RNA flow of cancer cell state has also been strengthened. I would note that this drug discovery part is still exploratory in nature, however, the analytical design and data mining is sound and, it will stimulate the field to contemplate these data and indications.

Overall, this is an exciting and novel work, with significant implications in the genomics of HNSCC.

We thank you for your kind comments and second review of our manuscript.

Comments on behalf of Reviewer #3

Major Comment

1) The Major Comment-1 from Reviewer-3 shares the similar concern with the Comment-5 from Reviewer-2, questioning the use of literature-curated genes in the annotation of tumor spots. During the revision, the authors have performed a completely new “de novo” data analysis which has addressed this question with satisfaction, validating also their initial observations and conclusions.

We thank you for your comments and second review of our manuscript.

2) The Major Comment-2 has been addressed by the authors.

3) For the Major Comment-3, the authors have now performed unsupervised clustering, which confirms their original supervised data. They have also provided explanations regarding a few samples showing the opposite correlation trends, which I agree.

We thank you for your comments and second review of our manuscript.

4) The Major Comment-4 has been addressed by the authors.

We thank you for your comments and second review of our manuscript.

5) Regarding the Major Comment-5, the authors have addressed this by identifying and quantifying the number of adjacent nonmalignant spots neighboring either malignant TC or LE spots.

We thank you for your comments and second review of our manuscript.

6) Addressing the Major Comment-5, the authors have now performed statistical quantification of the vector field effects.

We thank you for your comments and second review of our manuscript.

7) The authors have provided reasonable exploration and discussion to answer the Major Comment-7.

We thank you for your comments and second review of our manuscript.

8) Regarding the Major Comment-8, it is appropriate that the scope of such in silico drug discovery is restricted to HPV- OSCC for this current study.

We thank you for your comments and second review of our manuscript.

9) For Major Comment-9, the authors response is acceptable: some of the drugs identified have been previously evaluated in OSCC models using orthogonal approaches, and it is not imperative for the current study.

We thank you for your comments and second review of our manuscript.

Minor Comments

The authors have addressed all three of the Minor Comments.

We thank you for your comments and second review of our manuscript.

Reviewer #4 (Remarks to the Author):

All major issues raised have been addressed.

We thank you for your comment and second review of our manuscript.

Reviewer #5 (Remarks to the Author):

I have reviewed the revision submitted by the authors. In response to the first round of reviews the authors focussed their spatial transcriptomics analysis on 'cancer only' spots and I believe that this solves many of the problems from the original version. The authors have also added several important validation experiments that bolster the confidence in the results. The work has many interesting analyses and datasets that will be important for the community.

We thank you for your comment and second review of our manuscript.